# WERE RNNS ALL WE NEEDED?

## ABSTRACT

The introduction of Transformers in 2017 reshaped the landscape of deep learning. Originally proposed for sequence modelling, Transformers have since achieved widespread success across various domains. However, the scalability limitations of Transformers—particularly with respect to sequence length—have sparked renewed interest in novel recurrent models that are parallelizable during training, offer comparable performance, and scale more effectively. In this work, we revisit sequence modelling from a historical perspective, focusing on Recurrent Neural Networks (RNNs), which dominated the field for two decades before the rise of Transformers. Specifically, we examine LSTMs (1997) and GRUs (2014). We demonstrate that by simplifying these models, we can derive minimal versions (minLSTMs and minGRUs) that (1) use fewer parameters than their traditional counterparts, (2) are fully parallelizable during training, and (3) achieve surprisingly competitive performance on a range of tasks, rivalling recent models including Transformers.

## 1 INTRODUCTION

Since the 1990s, Recurrent Neural Networks (RNNs) (Elman, 1990), such as Long Short-Term Memory (LSTM) (Hochreiter & Schmidhuber, 1997) networks and later Gated Recurrent Units (GRUs) (Cho et al., 2014), have been go-to methods for sequence modelling tasks like machine translation and text generation. However, their inherently sequential nature, which limits parallelization, made these models computationally inefficient and too slow to train on long sequences, a common challenge in real-world applications.

In 2017, Transformers (Vaswani et al., 2017) revolutionized deep learning by introducing a parallelizable training mechanism through self-attention, achieving immediate success in sequence modelling. This breakthrough led to the development of popular large language models and quickly extended to other domains, including computer vision (Dosovitskiy et al., 2021), reinforcement learning (Chen et al., 2021), and bioinformatics (Jumper et al., 2021). However, while self-attention allows for efficient modelling of token-to-token interactions, it suffers from quadratic computational complexity, making Transformers prohibitively expensive for long sequences, especially in resource-constrained settings. To address this, numerous approaches have focused on improving Transformer efficiency, exploring ideas such as sparsity (Kitaev et al., 2019), low-rank approximations (Wang et al., 2020), and tiling (Dao et al., 2022).

Recently, the scalability limitations of Transformers have sparked renewed interest in alternative approaches: novel recurrent models that are parallelizable and scale more efficiently. Several promising methods have emerged in this space, including state-space models (Gu et al., 2021), linearized attention (Peng et al., 2023), and more recently, linear recurrent neural networks (Orvieto et al., 2023). Notably, these state-of-the-art recurrent models leverage input-dependent transitions and demonstrate strong performance similar to Transformers. These methods have shown success not only in scaling to large language models but also in extending to other domains, such as image (Zhu et al., 2024a) and graph-based data (Wang et al., 2024a).

In this work, we revisit sequence modelling from a historical perspective, focusing on the RNNs that dominated the field for two decades before the rise of Transformers. Specifically, we explore LSTMs (1997) and GRUs (2014), which are early examples of input-dependent recurrent models. We show that by removing the dependencies of their gates on previous states, we can train these models in parallel. Further simplification leads to minimal versions (minLSTMs and minGRUs)

that (1) use fewer parameters than their traditional counterparts, (2) are fully parallelizable during training, and (3) achieve surprisingly competitive performance on a range of tasks despite their simplicity, challenging the prevailing trend in the community toward increasing architectural and algorithmic complexity. In the appendix, we provide implementations of minGRU and minLSTM in plain PyTorch, with just a few lines of code, making these models lightweight and highly adaptable for beginners, practitioners, and researchers.

## 2 BACKGROUND

In this section, we review traditional recurrent neural networks (RNNs). RNNs are sequence models that maintain a hidden state across time steps, capturing temporal dependencies. As such, they are particularly well-suited for tasks involving sequential data, such as time series forecasting, natural language processing, and other tasks where context from previous steps informs current predictions. However, vanilla RNNs (Elman, 1990) face challenges related to vanishing and exploding gradients, which limit their ability to learn long-term dependencies.

### 2.1 LSTM

To address these issues, Hochreiter & Schmidhuber (1997) introduced Long Short-Term Memory (LSTM) networks. LSTMs are a highly successful type of RNN specifically designed to mitigate the vanishing gradient problem, enabling the model to effectively capture long-term dependencies. LSTMs are computed as follows:

$$
\begin{aligned}
\text{(Hidden State)} \quad & \boldsymbol{h}_t = \boldsymbol{o}_t \odot \tanh(\boldsymbol{c}_t) \\
\text{(Output Gate)} \quad & \boldsymbol{o}_t = \sigma(\text{Linear}_{d_h}([\boldsymbol{x}_t, \boldsymbol{h}_{t-1}])) \\
\text{(Cell State Recurrence)} \quad & \boldsymbol{c}_t = \boldsymbol{f}_t \odot \boldsymbol{c}_{t-1} + \boldsymbol{i}_t \odot \tilde{\boldsymbol{c}}_t \\
\text{(Forget Gate)} \quad & \boldsymbol{f}_t = \sigma(\text{Linear}_{d_h}([\boldsymbol{x}_t, \boldsymbol{h}_{t-1}])) \\
\text{(Input Gate)} \quad & \boldsymbol{i}_t = \sigma(\text{Linear}_{d_h}([\boldsymbol{x}_t, \boldsymbol{h}_{t-1}])) \\
\text{(Candidate Cell State)} \quad & \tilde{\boldsymbol{c}}_t = \tanh(\text{Linear}_{d_h}([\boldsymbol{x}_t, \boldsymbol{h}_{t-1}]))
\end{aligned}
$$

where $\odot$ denotes element-wise multiplication of vectors, $t$ is the current timestep, and $\boldsymbol{h}_t$ is the outputted hidden state. $[\boldsymbol{x}_t, \boldsymbol{h}_{t-1}]$ represents the concatenation of the input vector $\boldsymbol{x}_t$ at time step $t$ with the previous hidden state $\boldsymbol{h}_{t-1}$. $d_h$ denotes the size of the hidden state, while $\boldsymbol{c}_t$ is the cell state, which carries information across time steps, and $\tilde{\boldsymbol{c}}_t$ is the candidate cell state that will be added to the cell state.

The gates $\boldsymbol{i}_t$, $\boldsymbol{f}_t$, and $\boldsymbol{o}_t$ control the flow of information through the LSTM. The input gate $\boldsymbol{i}_t$ determines how much new information from the candidate cell state $\tilde{\boldsymbol{c}}_t$ should be added to the cell state $\boldsymbol{c}_t$. The forget gate $\boldsymbol{f}_t$ determines what portion of the previous cell state $\boldsymbol{c}_{t-1}$ should be discarded. The output gate $\boldsymbol{o}_t$ determines what information from the cell state should be output as the hidden state $\boldsymbol{h}_t$. The functions $\sigma$ (sigmoid) and $\tanh$ are used for scaling the values, ensuring that the outputs do not explode or vanish during training. An LSTM module maintains both a cell state and a hidden state, and, in total, contains $O(4d_h(d_x + d_h))$ parameters, where $d_x$ is the input size.

### 2.2 GRU

Simplifying LSTM, Cho et al. (2014) introduced the Gated Recurrent Unit (GRU), which uses only two gates and a single state (hidden state), in contrast to the LSTM's three gates and two states (hidden state and cell state). This reduced complexity allows GRUs to achieve faster training and inference times while still performing competitively on many tasks. GRUs are computed as follows:

$$
\begin{aligned}
\text{(Hidden State Recurrence)} \quad & \boldsymbol{h}_t = (\boldsymbol{1} - \boldsymbol{z}_t) \odot \boldsymbol{h}_{t-1} + \boldsymbol{z}_t \odot \tilde{\boldsymbol{h}}_t \\
\text{(Update Gate)} \quad & \boldsymbol{z}_t = \sigma(\text{Linear}_{d_h}([\boldsymbol{x}_t, \boldsymbol{h}_{t-1}])) \\
\text{(Reset Gate)} \quad & \boldsymbol{r}_t = \sigma(\text{Linear}_{d_h}([\boldsymbol{x}_t, \boldsymbol{h}_{t-1}])) \\
\text{(Candidate Hidden State)} \quad & \tilde{\boldsymbol{h}}_t = \tanh(\text{Linear}_{d_h}([\boldsymbol{x}_t, \boldsymbol{r}_t \odot \boldsymbol{h}_{t-1}]))
\end{aligned}
$$

where $\tilde{\boldsymbol{h}}_t$ represents the candidate hidden state, a potential new value for the hidden state. GRU combines the forget and input gates of LSTM into a single update gate, $\boldsymbol{z}_t \in (0,1)$, which determines how much of the past information should be carried forward (i.e., $1 - \boldsymbol{z}_t$) and how much new information from the candidate hidden state should be added (i.e., $\boldsymbol{z}_t$). Additionally, GRU removes LSTM's output gate and introduces a reset gate $\boldsymbol{r}_t$, which controls how much of the past hidden state $\boldsymbol{h}_{t-1}$ is used when computing the candidate hidden state $\tilde{\boldsymbol{h}}_t$.

By reducing the number of gates and states, GRU also decreases the total number of parameters and computations, requiring only $O(3d_h(d_x + d_h))$ parameters. However, both GRUs and LSTMs are still sequential-only models. As such, they require backpropagation through time (BPTT) during training, resulting in linear training time and limiting their ability to scale to long contexts.

### 2.3 PARALLEL SCAN

Due to this limitation, the introduction of Transformers in 2017 revolutionized the field by replacing LSTMs and GRUs as the de facto method for sequence modelling. Transformers leverage parallelization during training, overcoming the sequential bottleneck of traditional recurrent models. However, instead, Transformers have a quadratic complexity with respect to the sequence length, limiting their ability to scale to very long contexts, especially in resource-constrained settings.

In response, a resurgence of new recurrent sequence models has emerged, offering alternatives to Transformers. These models achieve comparable performance while being trainable in parallel and avoid the backpropagation through time (BPTT) issues that plagued traditional RNNs (e.g., LSTMs and GRUs). Among these innovations, many architectures rely on the parallel prefix scan algorithm (Blelloch, 1990) for efficient training.

The parallel scan algorithm is a parallel computation method for computing $N$ prefix computations from $N$ sequential data points via an associative operator $\oplus$ (e.g., $+$ and $\times$). The algorithm efficiently computes the sequence of prefix sums $\{\bigoplus_{i=1}^{k} u_i\}_{k=1}^{N}$ from the input sequence $\{u_k\}_{k=1}^{N}$. One important application of the parallel scan algorithm is in computing a popular class of recurrence relations of the form $v_t = a_t v_{t-1} + b_t$, where $v_t$, $a_t$, and $b_t$ are real numbers and $v_0 \leftarrow b_0$ (Martin & Cundy, 2018). This method takes as input the sequences $a_1, \ldots, a_n$ and $b_0, b_1, \ldots, b_n$, and computes the sequence $v_1, \ldots, v_n$ in parallel. This approach naturally extends to vector-valued recurrences, such as $\boldsymbol{v}_t = \boldsymbol{a}_t \odot \boldsymbol{v}_{t-1} + \boldsymbol{b}_t$, where $\odot$ denotes element-wise multiplication.

## 3 METHODOLOGY

Interestingly, we can see that the GRU's hidden state and LSTM's cell state recurrences resemble the vector formulation. In this section, we demonstrate that GRUs and LSTMs are trainable via parallel scan by removing their previous state dependencies from their various gates. Building on this, we further simplify these RNNs by removing their constraints on output range (i.e., $\tanh$). Combining the steps, we describe minimal versions of GRUs and LSTMs (minGRUs and minLSTMs) that are trainable in parallel.

### 3.1 A MINIMAL GRU: MINGRU

#### 3.1.1 STEP 1: DROP PREVIOUS STATE DEPENDENCIES FROM GATES

Revisiting GRU's hidden state recurrence which works as follows:

$$\boldsymbol{h}_t = (1 - \boldsymbol{z}_t) \odot \boldsymbol{h}_{t-1} + \boldsymbol{z}_t \odot \tilde{\boldsymbol{h}}_t$$

We can observe that the recurrence resembles the aforementioned parallel scan's formulation $\boldsymbol{v}_t = \boldsymbol{a}_t \odot \boldsymbol{v}_{t-1} + \boldsymbol{b}_t$ where $\boldsymbol{a}_t \leftarrow (1 - \boldsymbol{z}_t)$, $\boldsymbol{b}_t \leftarrow \boldsymbol{z}_t \odot \tilde{\boldsymbol{h}}_t$, and $\boldsymbol{v}_t \leftarrow \boldsymbol{h}_t$. However, $\boldsymbol{z}_t$ and $\tilde{\boldsymbol{h}}_t$ are dependent on the previous hidden state $\boldsymbol{h}_{t-1}$, i.e., $\boldsymbol{z}_t = \sigma(\text{Linear}_{d_h}([\boldsymbol{x}_t, \boldsymbol{h}_{t-1}]))$ and $\tilde{\boldsymbol{h}}_t = \tanh(\text{Linear}_{d_h}([\boldsymbol{x}_t, r_t \odot \boldsymbol{h}_{t-1}]))$. As a result, it is not possible to apply the parallel scan as is since the algorithm's inputs $\boldsymbol{a}_1, \ldots, \boldsymbol{a}_n$ and $\boldsymbol{b}_1, \ldots, \boldsymbol{b}_n$ are conditional on already knowing its outputs $\boldsymbol{h}_1, \ldots, \boldsymbol{h}_{n-1}$.

A simple remedy to this is to simplify GRU by removing their previous hidden state (i.e., $\boldsymbol{h}_{t-1}$) dependencies. Specifically, the changes are as follows:

$$\boldsymbol{z}_t = \sigma(\text{Linear}_{d_h}([\boldsymbol{x}_t, \boldsymbol{h}_{t-1}]))$$
$$\boldsymbol{r}_t = \sigma(\text{Linear}_{d_h}([\boldsymbol{x}_t, \boldsymbol{h}_{t-1}])) \qquad \Rightarrow \qquad \boldsymbol{z}_t = \sigma(\text{Linear}_{\boldsymbol{d}_h}(\boldsymbol{x}_t))$$
$$\tilde{\boldsymbol{h}}_t = \tanh(\text{Linear}_{d_h}([\boldsymbol{x}_t, \boldsymbol{r}_t \odot \boldsymbol{h}_{t-1}])) \qquad\qquad \tilde{\boldsymbol{h}}_t = \tanh(\text{Linear}_{\boldsymbol{d}_h}(\boldsymbol{x}_t))$$

By removing the dependence on $\boldsymbol{h}_{t-1}$ from the candidate hidden state $\tilde{\boldsymbol{h}}_t$, the reset gate $\boldsymbol{r}_t$ that would control $\boldsymbol{h}_{t-1}$ weight is also no longer needed and is removed. Without the dependencies on previous hidden states, the inputs to the algorithm $\boldsymbol{a}_1, \ldots, \boldsymbol{a}_n$ and $\boldsymbol{b}_1, \ldots, \boldsymbol{b}_n$ are all easily computed in parallel and can thus be used to compute $\boldsymbol{h}_1, \ldots, \boldsymbol{h}_n$ efficiently via the parallel scan.

Although there have been theoretical concerns about the absence of previous state dependencies (Merrill et al., 2024), there has also been substantial empirical evidence supporting the effectiveness of models that omit these dependencies, such as xLSTM (Beck et al., 2024) and Mamba (Gu & Dao, 2024). Instead of explicitly modelling dependencies on previous states to capture long-range dependencies, these kinds of recurrent models can learn them by stacking multiple layers. Notably, in the xLSTM paper, their fully parallelized version (xLSTM[1:0]), which eliminates hidden state dependencies, performed similarly to — and in some cases, better than — versions that retain these dependencies (e.g., xLSTM[7:1]).

### 3.1.2 STEP 2: DROP RANGE RESTRICTION OF CANDIDATE STATES

In GRU's hidden state recurrence, the proportion carried over from the previous hidden state $(\mathbf{1} - \boldsymbol{z}_t)$ and the amount added for the new candidate hidden state $(\boldsymbol{z}_t)$ sum to 1. As a result, the scale of GRU's hidden state value is time-independent. Instead, the scale of its hidden state depends on that of its candidate hidden states $\tilde{\boldsymbol{h}}_t$. The hyperbolic tangent function ($\tanh$) plays a crucial role in LSTMs and GRUs, restricting the range of (candidate) hidden states, i.e., $\tilde{\boldsymbol{h}}_t, \boldsymbol{h}_t \in (-1, 1)^{d_h}$. The $\tanh$ helps stabilize the training and mitigates vanishing gradients that result from applying sigmoid ($\sigma$) activations to linear transformations of the hidden state (e.g., $\boldsymbol{z}_t = \sigma(\text{Linear}_{d_h}([\boldsymbol{x}_t, \boldsymbol{h}_{t-1}]))$). In the previous step, these hidden state dependencies were removed. As such, we simplify GRU further by removing the range restriction ($\tanh$) on the (candidate) hidden states as follows:

$$\tilde{\boldsymbol{h}}_t = \tanh(\text{Linear}_{d_h}(\boldsymbol{x}_t)) \quad \Rightarrow \quad \tilde{\boldsymbol{h}}_t = \text{Linear}_{d_h}(\boldsymbol{x}_t)$$

### 3.1.3 MINGRU

Combining the two simplification steps results in a minimal version of GRU (minGRU):

| GRU |
|---|
| $\boldsymbol{h}_t = (\mathbf{1} - \boldsymbol{z}_t) \odot \boldsymbol{h}_{t-1} + \boldsymbol{z}_t \odot \tilde{\boldsymbol{h}}_t$ 
 $\boldsymbol{z}_t = \sigma(\text{Linear}_{d_h}([\boldsymbol{x}_t, \boldsymbol{h}_{t-1}]))$ 
 $\boldsymbol{r}_t = \sigma(\text{Linear}_{d_h}([\boldsymbol{x}_t, \boldsymbol{h}_{t-1}]))$ 
 $\tilde{\boldsymbol{h}}_t = \tanh(\text{Linear}_{d_h}([\boldsymbol{x}_t, \boldsymbol{r}_t \odot \boldsymbol{h}_{t-1}]))$ |

$\Rightarrow$

| minGRU |
|---|
| $\boldsymbol{h}_t = (\mathbf{1} - \boldsymbol{z}_t) \odot \boldsymbol{h}_{t-1} + \boldsymbol{z}_t \odot \tilde{\boldsymbol{h}}_t$ 
 $\boldsymbol{z}_t = \sigma(\text{Linear}_{d_h}(\boldsymbol{x}_t))$ 
 $\tilde{\boldsymbol{h}}_t = \text{Linear}_{d_h}(\boldsymbol{x}_t)$ |

The resulting model is significantly more efficient than the original GRU, requiring only $O(2 d_h d_x)$ parameters, compared to GRU's $O(3 d_h (d_x + d_h))$ parameters, where $d_x$ and $d_h$ denote the sizes of the input $x_t$ and the hidden state $h_t$, respectively. In RNNs, state expansion is often used (i.e., $d_h = \alpha d_x$, where $\alpha \geq 1$), which helps the models better capture features from the input data. minGRU uses approximately 33%, 22%, 17%, and 13% of the parameters of a GRU when $\alpha = 1, 2, 3, 4$, respectively.

Additionally, the minimal version of GRU can now be trained in parallel using the parallel scan algorithm, bypassing the need for backpropagation through time (BPTT). Pseudocode and a simple PyTorch implementation are included in the Appendix.

### 3.2 A MINIMAL LSTM: minLSTM

#### 3.2.1 STEP 1: DROP PREVIOUS STATE DEPENDENCIES FROM GATES

Revisiting LSTM's cell state recurrence which works as follows:

$$\boldsymbol{c}_t = \boldsymbol{f}_t \odot \boldsymbol{c}_{t-1} + \boldsymbol{i}_t \odot \tilde{\boldsymbol{c}}_t$$

Similar to GRU's hidden state, we can see that LSTM's cell state recurrence resembles the aforementioned parallel scan's formulation $\boldsymbol{v}_t = \boldsymbol{a}_t \odot \boldsymbol{v}_{t-1} + \boldsymbol{b}_t$ where $\boldsymbol{a}_t \leftarrow \boldsymbol{f}_t$, $\boldsymbol{b}_t \leftarrow \boldsymbol{i}_t \odot \tilde{\boldsymbol{c}}_t$, and $\boldsymbol{v}_t \leftarrow \boldsymbol{c}_t$. However, $\boldsymbol{f}_t$, $\boldsymbol{i}_t$ and $\tilde{\boldsymbol{c}}_t$ are dependent on the previous hidden state $\boldsymbol{h}_t$. As such, LSTM's cell state recurrence is unable to apply the parallel scan algorithm as is. We can address this in a similar fashion to GRU by removing their hidden state dependencies as follows:

$$
\begin{aligned}
\boldsymbol{f}_t &= \sigma(\text{Linear}_{d_h}([\boldsymbol{x}_t, \boldsymbol{h}_{t-1}])) & \boldsymbol{f}_t &= \sigma(\text{Linear}_{d_h}(\boldsymbol{x}_t)) \\
\boldsymbol{i}_t &= \sigma(\text{Linear}_{d_h}([\boldsymbol{x}_t, \boldsymbol{h}_{t-1}])) \quad \Rightarrow \quad & \boldsymbol{i}_t &= \sigma(\text{Linear}_{d_h}(\boldsymbol{x}_t)) \\
\tilde{\boldsymbol{c}}_t &= \tanh(\text{Linear}_{d_h}([\boldsymbol{x}_t, \boldsymbol{h}_{t-1}])) & \tilde{\boldsymbol{c}}_t &= \tanh(\text{Linear}_{d_h}(\boldsymbol{x}_t))
\end{aligned}
$$

#### 3.2.2 STEP 2: DROP RANGE RESTRICTION OF CANDIDATE STATES

Similar to GRUs, LSTMs leverage the hyperbolic tangent function ($\tanh$) to restrict the range of its states between $(-1, 1)$. LSTMs apply the range restriction twice: once when computing the candidate cell state and once when computing its hidden state. In this step, we drop both as follows:

$$
\begin{aligned}
\tilde{\boldsymbol{c}}_t &= \tanh(\text{Linear}_{d_h}(\boldsymbol{x}_t)) \quad \Rightarrow \quad & \tilde{\boldsymbol{c}}_t &= \text{Linear}_{d_h}(\boldsymbol{x}_t) \\
\boldsymbol{h}_t &= \boldsymbol{o}_t \odot \tanh(\boldsymbol{c}_t) & \boldsymbol{h}_t &= \boldsymbol{o}_t \odot \boldsymbol{c}_t
\end{aligned}
$$

#### 3.2.3 STEP 3: SIMPLIFYING SCALING OF OUTPUT

Continuing the trend of simplification, we drop the output gate $\boldsymbol{o}_t$ which scales the hidden state. Without the output gate, the normalized hidden state is equal to the cell state, i.e., $\boldsymbol{h}_t = \boldsymbol{o}_t \odot \boldsymbol{c}_t \Rightarrow \boldsymbol{h}_t = \boldsymbol{c}_t$, making having both a hidden and cell state unnecessary. As such, we drop the cell state as well, resulting in the following modification:

$$
\begin{aligned}
\boldsymbol{h}_t &= \boldsymbol{o}_t \odot \boldsymbol{c}_t \\
\boldsymbol{o}_t &= \sigma(\text{Linear}_{d_h}(\boldsymbol{x}_t)) \quad \Rightarrow \quad & \boldsymbol{h}_t &= \boldsymbol{f}_t \odot \boldsymbol{h}_{t-1} + \boldsymbol{i}_t \odot \tilde{\boldsymbol{h}}_t \\
\boldsymbol{c}_t &= \boldsymbol{f}_t \odot \boldsymbol{c}_{t-1} + \boldsymbol{i}_t \odot \tilde{\boldsymbol{c}}_t & \tilde{\boldsymbol{h}}_t &= \text{Linear}_{d_h}(\boldsymbol{x}_t) \\
\tilde{\boldsymbol{c}}_t &= \text{Linear}_{d_h}(\boldsymbol{x}_t)
\end{aligned}
$$

In many sequence modelling settings (e.g., text generation), the optimization objective/target is time-independent in scale. Recall LSTM's cell state recurrence $\boldsymbol{c}_t = \boldsymbol{f}_t \odot \boldsymbol{c}_{t-1} + \boldsymbol{i}_t \odot \tilde{\boldsymbol{c}}_t$ where $\boldsymbol{i}_t, \boldsymbol{f}_t \in (0, 1)^{d_h}$, and GRU's hidden state recurrence[1], $\boldsymbol{h}_t^{GRU} = (\boldsymbol{1} - \boldsymbol{z}_t) \odot \boldsymbol{h}_{t-1}^{GRU} + \boldsymbol{z}_t \odot \tilde{\boldsymbol{h}}_t^{GRU}$ where $\boldsymbol{z}_t \in (0, 1)^{d_h}$. GRUs retain $(\boldsymbol{1} - \boldsymbol{z}_t) \in (0, 1)$ of the previous hidden state and add $\boldsymbol{z}_t$ of the new candidate state. Since these proportions sum to $\boldsymbol{1}$, the model ensures its outputs (i.e., hidden states) are *time-independent* in scale. In contrast, LSTM's forget and input gates are computed independently (e.g., $\boldsymbol{f}_t, \boldsymbol{i}_t \to \boldsymbol{1}$ or $\boldsymbol{f}_t, \boldsymbol{i}_t \to \boldsymbol{0}$), making its states *time-dependent* in scale[2]. For tasks where time-independence is important, we can ensure LSTM's output is time-independent in scale by simply normalizing its input and forget gates, i.e., $\boldsymbol{f}_t', \boldsymbol{i}_t' \leftarrow \frac{\boldsymbol{f}_t}{\boldsymbol{f}_t + \boldsymbol{i}_t}, \frac{\boldsymbol{i}_t}{\boldsymbol{f}_t + \boldsymbol{i}_t}$, ensuring that $\boldsymbol{f}_t' + \boldsymbol{i}_t' = \boldsymbol{1}$ and the scale of LSTM's state is time-independent.

#### 3.2.4 minLSTM

Combining the three steps results in a minimal version of LSTM (minLSTM):

---

[1]A superscript is added to differentiate GRU's hidden state from LSTM's.

[2]For example, $\boldsymbol{c}_t \to \boldsymbol{c}_0 + \sum_{i=1}^{t} \tilde{\boldsymbol{c}}_t$ when $\boldsymbol{f}_{1:t}, \boldsymbol{i}_{1:t} \to 1$, growing in scale as the sequence length increases.

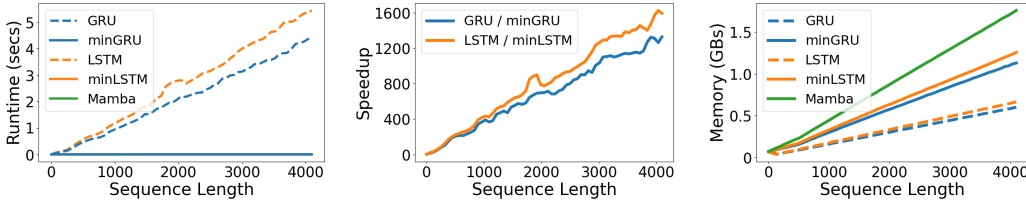

Figure 1: Training runtime (left), speedup (middle), and memory footprint (right) on a T4 GPU for a batch size of $64$. In the training runtime plot (left), minGRU, minLSTM, and Mamba lines overlap. These methods are approximately the same in training runtime.

**LSTM**

$$\boldsymbol{h}_t = \boldsymbol{o}_t \odot \tanh(\boldsymbol{c}_t)$$
$$\boldsymbol{o}_t = \sigma(\mathrm{Linear}_{d_h}([\boldsymbol{x}_t, \boldsymbol{h}_{t-1}]))$$
$$\boldsymbol{c}_t = \boldsymbol{f}_t \odot \boldsymbol{c}_{t-1} + \boldsymbol{i}_t \odot \tilde{\boldsymbol{c}}_t$$
$$\boldsymbol{f}_t = \sigma(\mathrm{Linear}_{d_h}([\boldsymbol{x}_t, \boldsymbol{h}_{t-1}]))$$
$$\boldsymbol{i}_t = \sigma(\mathrm{Linear}_{d_h}([\boldsymbol{x}_t, \boldsymbol{h}_{t-1}]))$$
$$\tilde{\boldsymbol{c}}_t = \tanh(\mathrm{Linear}_{d_h}([\boldsymbol{x}_t, \boldsymbol{h}_{t-1}]))$$

$\Rightarrow$

**minLSTM**

$$\boldsymbol{h}_t = \boldsymbol{f}_t \odot \boldsymbol{h}_{t-1} + \boldsymbol{i}_t \odot \tilde{\boldsymbol{h}}_t$$
$$\boldsymbol{f}_t = \sigma(\mathrm{Linear}_{d_h}(\boldsymbol{x}_t))$$
$$\boldsymbol{i}_t = \sigma(\mathrm{Linear}_{d_h}(\boldsymbol{x}_t))$$
$$\tilde{\boldsymbol{h}}_t = \mathrm{Linear}_{d_h}(\boldsymbol{x}_t)$$

where time-independent outputs can be achieved using a hidden state recurrence $\boldsymbol{h}_t = \boldsymbol{f}'_t \odot \boldsymbol{h}_{t-1} + \boldsymbol{i}'_t \odot \tilde{\boldsymbol{h}}_t$ with normalized forget $\boldsymbol{f}'_t$ and input $\boldsymbol{i}_t$ gates computed as $\boldsymbol{f}'_t, \boldsymbol{i}'_t \leftarrow \frac{\boldsymbol{f}_t}{\boldsymbol{f}_t + \boldsymbol{i}_t}, \frac{\boldsymbol{i}_t}{\boldsymbol{f}_t + \boldsymbol{i}_t}$.

The resulting model is significantly more efficient than the original LSTM, requiring only $O(3d_h d_x)$ parameters compared to LSTM's $O(4d_h(d_x + d_h))$. Considering state expansion (i.e., $d_h = \alpha d_x$, where $\alpha \geq 1$), minLSTM uses approximately $38\%, 25\%, 19\%$, or $15\%$ of the parameters of a LSTM when $\alpha = 1, 2, 3,$ or $4$ respectively.

Additionally, the minimal version of LSTM can now be trained in parallel using the parallel scan algorithm, bypassing the need for backpropagation through time (BPTT). Pseudocode and a simple PyTorch implementation are included in the Appendix.

## 4 WERE RNNs ALL WE NEEDED?

In this section, we compare the minimal versions (minLSTMs and minGRUs) with their traditional counterparts (LSTMs and GRUs) and modern sequence models. Pseudocode, PyTorch implementation, and detailed information regarding the experiment setup are available in the Appendix.

### 4.1 MINIMAL LSTMs AND GRUs ARE EFFICIENT

At test time, recurrent sequence models are typically rolled out sequentially, which makes inference relatively efficient. However, the main bottleneck for traditional RNNs lies in their training, which requires linear time due to backpropagation through time (BPTT). This computational inefficiency contributed to the eventual deprecation of many earlier RNN-based models.

In this section, we compare the resource requirements for training traditional RNNs (LSTM and GRU), their simplified counterparts (minLSTM and minGRU)[3], and Mamba (using the official implementation), a recent popular recurrent sequence model.

For these experiments, a fixed batch size of 64 was used while varying the sequence length. We measure both the total runtime and memory complexity involved in performing a forward pass,

---

[3]See Appendix for the PyTorch implementations of minLSTM and minGRU written in a few lines.

computing the loss, and performing backpropagation to compute gradients. To ensure a fair and direct comparison, all models were tested with the same number of layers.

**Runtime.** We would like to highlight that inference speed can vary depending on hardware and implementation. PyTorch's built-in RNNs are highly optimized low-level GPU implementations. For a more fair comparison, in these experiments, minGRU, minLSTM, GRU, and LSTM were all written in plain Pytorch. In terms of runtime (see Figure 1 (left)), the simplified versions of LSTM and GRU (minLSTM and minGRU) Mamba achieve similar runtimes. Averaging over 100 runs, the runtime for sequence lengths of 512 for minLSTM, minGRU, and Mamba were 2.97, 2.72, and 2.71 milliseconds respectively. For a sequence with length 4096, the runtime were 3.41, 3.25, and 3.15 respectively. In contrast, the traditional RNN counterparts (LSTMs and GRUs) required a runtime that scaled linearly with respect to sequence length. For a sequence length of 512, minGRUs and minLSTMs were $175\times$ and $235\times$ faster per training step (see Figure 1 (middle)) than GRUs and LSTMs on a T4 GPU. The improvement is even more significant as sequences grow in length with minGRUs and minLSTMs being $1324\times$ and $1361\times$ faster for a sequence length of 4096. As such, in a setting where minGRU would take a day to finish training for a fixed number of epochs, its traditional counterpart GRU could take over 3 years.

**Memory.** By leveraging a parallel scan algorithm to compute the outputs in parallel efficiently, minGRU, minLSTM, and Mamba create a larger computational graph, thus needing more memory compared to traditional RNNs (see Figure 1 (right)). The minimal variants (minGRU and minL-STM) use $\sim 88\%$ more memory compared to their traditional counterparts (GRU and LSTM). Mamba uses $56\%$ more memory compared to minGRU. In practice, however, runtime is often the bottleneck when training RNNs.

**Effect of removing $h_{t-1}$.** The original LSTM and GRU compute their various gates using their inputs $x_t$ and previous hidden states $h_{t-1}$. These models leverage their time-dependent gates to learn complex functions. However, minLSTM and minGRU's training efficiencies are achieved by dropping their gates' dependencies on the previous hidden states $h_{t-1}$. As a result, minLSTM and minGRU's gates are dependent only on their inputs $x_t$, resulting in a simpler recurrent module. As such, the gates of a model consisting of a single layer of minLSTM or minGRU are *time-independent* due to being conditioned on *time-independent* inputs $x_{1:n}^{(1)}$.

However, in deep learning, models are constructed by stacking modules. Although the inputs to the first layer $x_{1:n}^{(1)}$ is *time-independent*, its outputs $h_{1:n}^{(1)}$ are *time-dependent* and are used as the inputs to the second layer, i.e., $x_{1:n}^{(2)} \leftarrow h_{1:n}^{(1)}$. As such, beginning from the second layer onwards, minLSTM and minGRU's gates will also be time-dependent, resulting in the modelling of more complex functions. In Table 1, we compare the performance of the models with varying numbers of layers on the Selective Copying Task from the Mamba paper (Gu & Dao, 2024). We can immediately see the impact of the time dependencies: increasing the number of layers to 2 or more drastically increases performance.

| Model | # Layers | Accuracy |
|---------|----------|----------------|
| MinLSTM | 1 | $37.6 \pm 2.0$ |
| | 2 | $85.7 \pm 5.8$ |
| | 3 | $96.0 \pm 2.8$ |
| MinGRU | 1 | $37.0 \pm 2.3$ |
| | 2 | $96.8 \pm 3.2$ |
| | 3 | $99.5 \pm 0.2$ |

Table 1: Comparison of the number of layers on the Selective Copying Task (Gu & Dao, 2024).

**Training Stability.** Another effect of the number of layers is increased stability with decreased variance in the accuracy as the number of layers increases (see Table 1). Furthermore, although minLSTM and minGRU both solve the Selective Copying task, we can see that minGRU is an empirically more stable method than minLSTM, solving the task with more consistency and lower variance. minLSTM discards old information and adds new information, controlling the ratio with two sets of parameters (forget and input gate). During training, the two sets of parameters are tuned in different directions, making the ratio harder to control and optimize. In contrast, minGRU's discarding and adding of information is controlled by a single set of parameters (update gate).

## 4.2 MINIMAL RNNS PERFORM SURPRISINGLY WELL

| Dataset | DT | DS4 | DAaren | DMamba | minLSTM | minGRU |
|---|---|---|---|---|---|---|
| HalfCheetah-M | 42.6 | 42.5 | 42.2 | 42.8 | 42.7 ± 0.7 | 43.0 ± 0.4 |
| Hopper-M | 68.4 | 54.2 | 80.9 | 83.5 | 85.0 ± 4.4 | 79.4 ± 8.2 |
| Walker-M | 75.5 | 78.0 | 74.4 | 78.2 | 72.0 ± 7.5 | 73.3 ± 3.3 |
| HalfCheetah-M-R | 37.0 | 15.2 | 37.9 | 39.6 | 38.6 ± 1.1 | 38.5 ± 1.1 |
| Hopper-M-R | 85.6 | 49.6 | 77.9 | 82.6 | 88.5 ± 4.7 | 90.5 ± 0.9 |
| Walker-M-R | 71.2 | 69.0 | 71.4 | 70.9 | 69.7 ± 10.7 | 72.8 ± 8.9 |
| HalfCheetah-M-E | 88.8 | 92.7 | 75.7 | 91.9 | 85.4 ± 1.7 | 86.3 ± 0.5 |
| Hopper-M-E | 109.6 | 110.8 | 103.9 | 111.1 | 110.3 ± 1.6 | 109.7 ± 2.7 |
| Walker-M-E | 109.3 | 105.7 | 110.5 | 108.3 | 110.3 ± 0.5 | 110.3 ± 0.4 |
| Average | 76.4 | 68.6 | 75.0 | 78.8 | 78.1 | 78.2 |

Table 3: Reinforcement Learning results on the D4RL (Fu et al., 2020) datasets. We report the expert normalized returns (higher is better), following (Fu et al., 2020), averaged across five random seeds. The minimal versions of LSTM and GRU, minLSTM and minGRU outperform Decision S4 (David et al., 2023) and perform comparably with Decision Mamba (Ota, 2024), (Decision) Aaren (Feng et al., 2024) and Decision Transformer (Chen et al., 2021).

In this section, we focus on the empirical performance of these minimal versions of decades-old models LSTMs (1997) and GRUs (2014), comparing them to several modern sequence models. It is important to note that the primary goal of our work is not to attain the best performance on specific tasks but to demonstrate that simplifying traditional architectures can yield competitive results, comparable to those of recent sequence models.

**Selective Copy.** We begin by considering the Selective Copying task, originally introduced in the influential Mamba paper (Gu & Dao, 2024). This task served as a key benchmark that demonstrated the improvements made by Mamba's state-space model, S6, over previous state-of-the-art models such as S4 (Gu et al., 2021) and Hyena (Poli et al., 2023). The task requires models to perform content-aware reasoning, where they must selectively memorize relevant tokens while filtering out irrelevant ones.

In Table 2, we compare the simplified versions of LSTMs and GRUs (minLSTM and minGRU) with several well-known recurrent sequence models that can be trained in parallel, including S4 (Gu et al., 2021), H3 (Fu et al., 2023), Hyena (Poli et al., 2023), and Mamba (S6) (Gu & Dao, 2024). The results for these baselines are directly quoted from the Mamba paper. Among these, only Mamba's S6 model succeeds in solving the task.

Both minGRU and minLSTM are able to solve the Selective Copying task as well, achieving performance comparable to S6 and surpassing the other modern baselines, highlighting the effectiveness of these traditional models LSTMs and GRUs, which utilize content-aware gating mechanisms.

| Model | Layer | Accuracy |
|---|---|---|
| H3 | Hyena | 30.1 |
| Mamba | Hyena | 28.4 |
| S4 | S4 | 18.3 |
| H3 | S4 | 57.0 |
| Mamba | S4 | 56.4 |
| S4 | S6 | 97.0 |
| H3 | S6 | 99.7 |
| Mamba | S6 | 99.8 |
| minGRU | minGRU | 99.5 ± 0.2 |
| minLSTM | minLSTM | 96.0 ± 2.8 |

Table 2: Selective Copy Task. minLSTM, minGRU, and Mamba's S6 (Gu & Dao, 2024) are capable of solving this task. Other methods such as S4, H3, and Hyena at best only partially solve the task.

**Reinforcement Learning.** Next, we consider the MuJoCo locomotion tasks from the D4RL benchmark (Fu et al., 2020). Specifically, we consider the three environments: HalfCheetah, Hopper, and Walker. For each environment, the models are trained on three datasets of varying data quality: Medium (M), Medium-Replay (M-R), and Medium-Expert (M-E).

In Table 3, we compare minLSTM and minGRU with various Decision Transformer variants, including the original Decision Transformer (DT) (Chen et al., 2021), Decision S4 (DS4) (David et al., 2023), Decision Mamba (Ota, 2024), and (Decision) Aaren (Feng et al., 2024). The baseline results are retrieved from the Decision Mamba and Aaren papers. minLSTM and minGRU

outperform Decision S4 and achieve performance competitive with Decision Transformer, Aaren, and Mamba. Unlike other recurrent methods, Decision S4 is a model whose recurrence transitions are not input-aware, affecting their performance. In terms of average score across the $3 \times 3 = 9$ datasets, minLSTM and minGRU outperform all the baselines except for Decision Mamba where the difference is marginal.

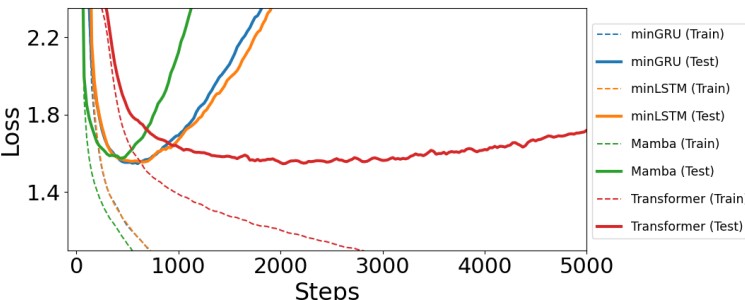

Figure 2: Language Modelling results on the Shakespeare dataset. Minimal versions of decade-old RNNs (LSTMs and GRUs) performed comparably to Mamba and Transformers. Transformers required $\sim 2.5\times$ more training steps to achieve comparable performance, overfitting eventually.

**Language Modelling.** Finally, we consider a language modelling task. In this setting, we train a character-level GPT on the works of Shakespeare using the nanoGPT (Karpathy, 2022) framework. In Figure 2, we plot the learning curves with a cross-entropy loss comparing the proposed minimal LSTM and GRU (minLSTM and minGRU) with Mamba and Transformers. We found that minGRU, minLSTM, Mamba, and Transformers achieved comparable test losses of $1.548$, $1.555$, $1.575$, and $1.547$ respectively. Mamba performed slightly worse than the other models but trained faster, particularly in the early stages, achieving its best performance at $400$ steps while minGRU and minLSTM continued training until $575$ and $625$ steps respectively. In contrast, Transformers trained significantly slower, requiring $2000$ steps ($\sim 2.5\times$) more training steps than minGRU to achieve comparable performance, making it significantly slower and more resource-intensive to train (quadratic complexity compared to minGRU, minLSTM, and Mamba's linear complexity).

## 5 RELATED WORK

In this section, we provide a brief overview of recent efficient recurrent sequence models that have demonstrated strong empirical performance, rivalling Transformers, while offering better scalability. For a more comprehensive discussion on the resurgence of efficient recurrent models, we refer the reader to recent surveys (Tiezzi et al., 2024). Broadly speaking, these models have evolved in three key directions:

**(Deep) State-Space Models (SSMs).** Building on continuous-time linear systems, Gu et al. (2021) introduced S4, a state-space model that can be unrolled like an RNN during inference and trained similarly to a convolutional neural network. S4's success paved the way for numerous subsequent developments in the field (Gu et al., 2022; Gupta et al., 2022; Hasani et al., 2023; Smith et al., 2023) and their applications across various domains such as language processing (Mehta et al., 2023) and audio analysis (Goel et al., 2022). More recently, Mamba emerged as a significant breakthrough in SSMs, surpassing previous models and attracting considerable attention. One of the key innovations in Mamba was the introduction of S6, a state-space model with input-dependent transition matrices, contrasting with earlier models that used input-independent transition matrices. The success of Mamba and other state-space models has led to the publication of several comprehensive surveys on the topic (Wang et al., 2024b; Patro & Agneeswaran, 2024; Qu et al., 2024).

**Recurrent Versions of Attention.** Another popular direction is that of attention, specifically related to linear attention (Katharopoulos et al., 2020). For example, Sun et al. (2023) and Qin et al. (2023) introduced linear attention models that use an input-independent gating mechanism (decay factor). In contrast, Katsch (2023) and Yang et al. (2024) proposed linear attention variants that use input-dependent gating. Recently, Feng et al. (2024) showed that softmax attention can also be viewed as an RNN and proposed a recurrent model based on their RNN formulation.

**Parallelizable RNNs.** Our work is closely related to several notable papers that parallelize RNNs. Bradbury et al. (2017) modified classical gated RNNs to leverage convolutional layers for efficiency, applying them temporally. Martin & Cundy (2018) demonstrated that RNNs with linear dependencies can be efficiently trained using a parallel scan. Building on this work, the authors introduced GILR, a gated linear RNN, where the outputs can serve as a surrogate for the previous state dependencies in traditional RNNs (e.g., LSTMs), enabling parallel training. Notably, minGRU is equivalent to GILR but without an activation function. More recently, Orvieto et al. (2023) proposed a linear gated RNN that leverages complex diagonal recurrences and an exponential parameterization, achieving comparable performance to state-space models. Qin et al. (2024b) introduced HGRN whose token mixer HGRU is a linear gated RNN augmented with complex value (polar coordinate) recurrences, lower bounds on their forget gate, and an output gate. HGRN2 (Qin et al., 2024a) improved HGRN by incorporating state expansion. Beck et al. (2024) extends LSTM using exponential gating and a normalizer state. Their xLSTM consists of parallelizable (mLSTM) and sequential-only (sLSTM) versions. mLSTM removes the hidden state dependencies to enable parallelization, introduces a matrix memory cell, and uses a query vector for retrieval from the memory. Zhu et al. (2024b) builds on insights from HGRN and revisits GRUs, introducing a parallelizable token mixer that removes matrix multiplications and leverages ternary weight quantization.

## 6 CONCLUSION

In this work, we revisited the history of sequence modelling, focusing on traditional RNNs, specifically LSTMs (1997) and GRUs (2014), which dominated the field for two decades before the rise of Transformer models. We demonstrated that we can enable parallel training of traditional RNNs by removing their gates' dependencies on previous states. Further simplification of these architectures led to minimal versions—minLSTMs and minGRUs—which offer several advantages: (1) fewer parameters than their traditional counterparts, (2) full parallelizability during training, and (3) surprisingly competitive performance across a range of tasks, rivalling modern models despite their simplicity. In the appendix, we provide implementations of minGRU and minLSTM in plain PyTorch, requiring only a few lines of code. This makes them lightweight and accessible to beginners, practitioners, and researchers alike. We hope this work sparks a broader conversation about the evolution of sequence modelling, encouraging a reevaluation of simpler foundational models like LSTM and GRU in light of newer, more complex architectures. Given the surprising effectiveness of these minimal versions of decades-old RNNs, alongside the recent success of modern RNN architectures, we pose the question: "Were RNNs all we needed?"

### LIMITATIONS

Modern models such as Mamba and xLSTM were run on modern A100 GPUs with 80 GB of memory. In contrast, our experiments were conducted on older GPUs (i.e., P100, T4, and Quadro 5000) with only 16 GB of memory (roughly 20% of the memory available to the other models). These hardware constraints impacted our ability to perform large-scale experiments. To accommodate the memory limitations, we used gradient accumulation for training some tasks, reducing the effective batch size by half, which resulted in significantly slower training times. While this approach allowed us to run experiments within the available memory constraints, it also limited the scale of our evaluations.

Despite these limitations, we believe that the conclusions drawn from our experiments are likely to generalize to larger-scale settings. The minimal versions of traditional RNNs share fundamental similarities with many recent recurrent sequence models (e.g., input-dependent gating), which suggests that their performance would likely be consistent on larger datasets given sufficient computational resources.

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

# A IMPLEMENTATION DETAILS: VANILLA VERSION

In this section, we provide the pseudocode and equivalent PyTorch code for minGRU and minL-STM. When performing repeated multiplications such as in many recurrent sequence models, numerical instabilities are common, especially during training. As such, we trained using a log-space implementation (see Section B) for improved numerical stability.

## A.1 PSEUDOCODE: VANILLA VERSION

### A.1.1 MINGRU: A MINIMAL GRU

---

**Algorithm 1** Sequential Mode: Minimal Version of GRU (minGRU)

---

**Input:** $\boldsymbol{x}_t, \boldsymbol{h}_{t-1}$
**Output:** $\boldsymbol{h}_t$
    $\boldsymbol{z}_t \leftarrow \sigma(\text{Linear}_{d_h}(\boldsymbol{x}_t))$
    $\tilde{\boldsymbol{h}}_t \leftarrow \text{Linear}_{d_h}(\boldsymbol{x}_t)$
    $\boldsymbol{h}_t \leftarrow (\boldsymbol{1} - \boldsymbol{z}_t) \odot \boldsymbol{h}_{t-1} + \boldsymbol{z}_t \odot \tilde{\boldsymbol{h}}_t$

---

---

**Algorithm 2** Parallel Mode: Minimal Version of GRU (minGRU)

---

**Input:** $\boldsymbol{x}_{1:t}, \boldsymbol{h}_0$
**Output:** $\boldsymbol{h}_{1:t}$
    $\boldsymbol{z}_{1:t} \leftarrow \sigma(\text{Linear}_{d_h}(\boldsymbol{x}_{1:t}))$
    $\tilde{\boldsymbol{h}}_{1:t} \leftarrow \text{Linear}_{d_h}(\boldsymbol{x}_{1:t})$
    $\boldsymbol{h}_{1:t} \leftarrow \text{ParallelScan}((\boldsymbol{1} - \boldsymbol{z}_{1:t}), [\boldsymbol{h}_0, \boldsymbol{z}_{1:t} \odot \tilde{\boldsymbol{h}}_{1:t}])$

---

### A.1.2 MINLSTM: A MINIMAL LSTM

---

**Algorithm 3** Sequential Mode: Minimal Version of LSTM (minLSTM) with length independence scaling

---

**Input:** $\boldsymbol{x}_t, \boldsymbol{h}_{t-1}$
**Output:** $\boldsymbol{h}_t$
    $\boldsymbol{f}_t \leftarrow \sigma(\text{Linear}_{d_h}(\boldsymbol{x}_t))$
    $\boldsymbol{i}_t \leftarrow \sigma(\text{Linear}_{d_h}(\boldsymbol{x}_t))$
    $\boldsymbol{f}'_t, \boldsymbol{i}'_t \leftarrow \frac{\boldsymbol{f}_t}{\boldsymbol{f}_t+\boldsymbol{i}_t}, \frac{\boldsymbol{i}_t}{\boldsymbol{f}_t+\boldsymbol{i}_t}$
    $\tilde{\boldsymbol{h}}_t \leftarrow \text{Linear}_{d_h}(\boldsymbol{x}_t)$
    $\boldsymbol{h}_t \leftarrow \boldsymbol{f}'_t \odot \boldsymbol{h}_{t-1} + \boldsymbol{i}'_t \odot \tilde{\boldsymbol{h}}_t$

---

---

**Algorithm 4** Parallel Mode: Minimal Version of LSTM (minLSTM) with length independence scaling

---

**Input:** $\boldsymbol{x}_{1:t}, \boldsymbol{h}_0$
**Output:** $\boldsymbol{h}_{1:t}$
    $\boldsymbol{f}_{1:t} \leftarrow \sigma(\text{Linear}_{d_h}(\boldsymbol{x}_{1:t}))$
    $\boldsymbol{i}_{1:t} \leftarrow \sigma(\text{Linear}_{d_h}(\boldsymbol{x}_{1:t}))$
    $\boldsymbol{f}'_{1:t}, \boldsymbol{i}'_{1:t} \leftarrow \frac{\boldsymbol{f}_{1:t}}{\boldsymbol{f}_{1:t}+\boldsymbol{i}_{1:t}}, \frac{\boldsymbol{i}_{1:t}}{\boldsymbol{f}_{1:t}+\boldsymbol{i}_{1:t}}$
    $\tilde{\boldsymbol{h}}_{1:t} \leftarrow \text{Linear}_{d_h}(\boldsymbol{x}_{1:t})$
    $\boldsymbol{h}_{1:t} \leftarrow \text{ParallelScan}(\boldsymbol{f}'_{1:t}, [\boldsymbol{h}_0, \boldsymbol{i}'_{1:t} \odot \tilde{\boldsymbol{h}}_{1:t}])$

---

## A.2 PyTorch Code: Vanilla Version

### A.2.1 minGRU: A Minimal GRU

```python
def forward(self, x_t, h_prev):
    # x_t: (batch_size, 1, input_size)
    # h_prev: (batch_size, 1, hidden_size)

    z_t = torch.sigmoid(self.linear_z(x_t))
    h_tilde = self.linear_h(x_t)
    h_t = (1 - z_t) * h_prev + z_t * h_tilde
    return h_t
```

Listing 1: Sequential Mode: Minimal Version of GRU (minGRU)

```python
def forward(self, x, h_0):
    # x: (batch_size, seq_len, input_size)
    # h_0: (batch_size, 1, hidden_size)

    z = torch.sigmoid(self.linear_z(x))
    h_tilde = self.linear_h(x)
    h = parallel_scan((1 - z),
        torch.cat([h_0, z * tilde_h], dim=1))
    return h
```

Listing 2: Parallel Mode: Minimal Version of GRU (minGRU)

### A.2.2 minLSTM: A Minimal LSTM

```python
def forward(self, x_t, h_prev):
    # x_t: (batch_size, 1, input_size)
    # h_prev: (batch_size, 1, hidden_size)

    f_t = torch.sigmoid(self.linear_f(x_t))
    i_t = torch.sigmoid(self.linear_i(x_t))
    tilde_h_t = self.linear_h(x_t)
    f_prime_t = f_t / (f_t + i_t)
    i_prime_t = i_t / (f_t + i_t)
    h_t = f_prime_t * h_prev + i_prime_t * tilde_h_t
    return h_t
```

Listing 3: Sequential Mode: Minimal Version of LSTM (minLSTM) with length independence scaling

```python
def forward(self, x, h_0):
    # x: (batch_size, seq_len, input_size)
    # h_0: (batch_size, 1, hidden_size)

    f = torch.sigmoid(self.linear_f(x))
    i = torch.sigmoid(self.linear_i(x))
    tilde_h = self.linear_h(x)
    f_prime = f / (f + i)
    i_prime = i / (f + i)
    h = parallel_scan(f_prime,
        torch.cat([h_0, i_prime * tilde_h], dim=1))
    return h
```

Listing 4: Parallel Mode: Minimal Version of LSTM (minLSTM) with length independence scaling

# B   IMPLEMENTATION DETAILS: LOG-SPACE VERSION (ADDITIONAL NUMERICAL STABILITY)

In this section, we detail the log-space version of minLSTM and minGRU for improved numerical stability. During training, the parallel modes are used to avoid backpropagation through time (BPTT), speeding up the training time significantly. At inference time, the sequential modes are used.

## B.1   PARALLEL SCAN: LOG-SPACE IMPLEMENTATION

Recall that, the parallel scan's objective is to compute $h_{1:t}$ where $h_k = a_k \odot h_{k-1} + b_k$. In code, the vanilla parallel scan function would take as input: coefficients $a_{1:t}$ and values $b_{0:t}$. The function then outputs $h_{1:t}$. For numerical stability, we consider a log-space implementation which takes as input $\log(a_{1:t})$ and $\log(b_{0:t})$ instead and outputs $h_{1:t}$. The code for the parallel scan in log-space is included below and is based on the code by Heinsen (2023).

```python
def parallel_scan_log(log_coeffs, log_values):
    # log_coeffs: (batch_size, seq_len, input_size)
    # log_values: (batch_size, seq_len + 1, input_size)
    a_star = F.pad(torch.cumsum(log_coeffs, dim=1), (0, 0, 1, 0))
    log_h0_plus_b_star = torch.logcumsumexp(
                                    log_values - a_star, dim=1)
    log_h = a_star + log_h0_plus_b_star
    return torch.exp(log_h)[:, 1:]
```

Listing 5: Parallel scan based on Heinsen (2023). This function computes $h_{1:t}$ given log coefficients $\log(a_{1:t})$ and log values $\log(b_{0:t})$.

## B.2   PSEUDOCODE: LOG-SPACE VERSION

For maximal numerical stability, we rewrite the log-space versions of minGRU and minLSTM.

### B.2.1   MINGRU: A MINIMAL GRU

Recall minGRU's recurrence is as follows $h_t \leftarrow (1 - z_t) \odot h_{t-1} + z_t \odot \tilde{h}_t$. As such, $a_t \leftarrow (1 - z_t)$ and $b_t \leftarrow z_t \odot \tilde{h}_t$ where $z_t = \sigma(k_t)$ and $k_t = \text{Linear}_{d_h}(x_t)$. As a result, $\log(a_t) \leftarrow \log(1 - z_t)$ and $\log(b_t) \leftarrow \log(z_t) + \log(\tilde{h})_t$. We can break down these down as follows:

$$\log(z_t) = \log(\sigma(k_t))$$
$$= \log\left(\frac{1}{1 + \exp(-k_t)}\right)$$
$$= -\text{Softplus}(-k_t)$$
$$\log(1 - z_t) = \log\left(\frac{\exp(-k_t)}{1 + \exp(-k_t)}\right)$$
$$= \log\left(\frac{1}{1 + \exp(k_t)}\right)$$
$$= -\text{Softplus}(k_t)$$

where $k_t = \text{Linear}_{d_h}(x_t)$. However, we need to compute $\log(\tilde{h})_t$ which is inconvenient if $\tilde{h}_t$ has some negative values. We could use complex numbers and a complex number version of the parallel scan, but this would result in the parallel scan increasing in complexity. Instead, we propose to ensure that $\tilde{h}_t > 0$. This be can done in a variety of ways. In our experiments, we added a continuous activation function $g$ replacing $\tilde{h}_t \leftarrow \text{Linear}_{d_h}(x_t)$ with $\tilde{h}_t \leftarrow g(\text{Linear}_{d_h}(x_t))$ where
$$g(x) = \begin{cases} x + 0.5, & \text{if } x \geq 0 \\ \sigma(x), & \text{otherwise} \end{cases} \text{ and its log: } \log(g(x)) = \begin{cases} \log(x + 0.5), & \text{if } x \geq 0 \\ -\text{Softplus}(-x), & \text{otherwise} \end{cases}.$$

At inference time, the sequential mode (Algorithm 5) is used. During training, the parallel mode (Algorithm 6) is used.

---

**Algorithm 5** Sequential Mode: Minimal Version of GRU (minGRU) trained in log-space

---

**Input:** $\boldsymbol{x}_t, \boldsymbol{h}_{t-1}$
**Output:** $\boldsymbol{h}_t$
   $\boldsymbol{z}_t \leftarrow \sigma(\text{Linear}_{d_h}(\boldsymbol{x}_t))$
   $\tilde{\boldsymbol{h}}_t \leftarrow \text{g}(\text{Linear}_{d_h}(\boldsymbol{x}_t))$
   $\boldsymbol{h}_t \leftarrow (\boldsymbol{1} - \boldsymbol{z}_t) \odot \boldsymbol{h}_{t-1} + \boldsymbol{z}_t \odot \tilde{\boldsymbol{h}}_t$

---

**Algorithm 6** Parallel Mode: Minimal Version of GRU (minGRU) for training in log-space

---

**Input:** $\boldsymbol{x}_{1:t}, \boldsymbol{h}_0$
**Output:** $\boldsymbol{h}_{1:t}$
   $\text{linear\_z} \leftarrow \text{Linear}_{d_h}$
   $\log\_\boldsymbol{z}_{1:t} \leftarrow -\text{Softplus}(\text{linear\_z}(-\boldsymbol{x}_{1:t}))$
   $\text{log\_coeffs} \leftarrow -\text{Softplus}(\text{linear\_z}(\boldsymbol{x}_{1:t}))$
   $\log\_\boldsymbol{h}_0 \leftarrow \log(\boldsymbol{h}_0)$
   $\log\_\tilde{\boldsymbol{h}}_{1:t} \leftarrow \text{log\_g}(\text{Linear}_{d_h}(\boldsymbol{x}_{1:t}))$
   $\boldsymbol{h}_{1:t} \leftarrow \text{ParallelScanLog}(\text{log\_coeffs}, [\log\_\boldsymbol{h}_0, \log\_\boldsymbol{z}_{1:t} + \log\_\tilde{\boldsymbol{h}}_{1:t})$

---

### B.2.2 MINLSTM: A MINIMAL LSTM

We also derive minLSTM's (with length independence scaling) log-space formulation as well. Recall minLSTM's (with length independence scaling) recurrence is as follows $\boldsymbol{h}_t \leftarrow \boldsymbol{f}'_t \odot \boldsymbol{h}_{t-1} + \boldsymbol{i}'_t \odot \tilde{\boldsymbol{h}}_t$. As such, $\boldsymbol{a}_t \leftarrow \boldsymbol{f}'_t$ and $\boldsymbol{b}_t \leftarrow \boldsymbol{i}'_t \odot \tilde{\boldsymbol{h}}_t$. As a result, $\log(\boldsymbol{a}_t) \leftarrow \log(\boldsymbol{f}'_t)$ and $\log(\boldsymbol{b}_t) \leftarrow \log(\boldsymbol{i}'_t) + \log(\tilde{\boldsymbol{h}}_t)$.

$$
\begin{aligned}
\log(\boldsymbol{f}'_t) &= \log\left(\frac{\boldsymbol{f}_t}{\boldsymbol{f}_t + \boldsymbol{i}_t}\right) \\
&= \log\left(\frac{1}{1 + \frac{\boldsymbol{i}_t}{\boldsymbol{f}_t}}\right) \\
&= -\log\left(1 + \frac{\boldsymbol{i}_t}{\boldsymbol{f}_t}\right) \\
&= -\log\left(1 + \exp\left(\log\left(\frac{\boldsymbol{i}_t}{\boldsymbol{f}_t}\right)\right)\right) \\
&= -\text{Softplus}\left(\log\left(\frac{\boldsymbol{i}_t}{\boldsymbol{f}_t}\right)\right) \\
&= -\text{Softplus}\left(\log(\boldsymbol{i}_t) - \log(\boldsymbol{f}_t)\right)
\end{aligned}
$$

Recall that $\boldsymbol{i}_t$ and $\boldsymbol{f}_t$ are computed via sigmoid. In other words, $\boldsymbol{i}_t = \sigma(\boldsymbol{k}_t)$ and $\boldsymbol{f}_t = \sigma(\boldsymbol{p}_t)$ where $\boldsymbol{k}_t = \text{Linear}_{d_h}(\boldsymbol{x}_t)$ and $\boldsymbol{p}_t = \text{Linear}_{d_h}(\boldsymbol{x}_t)$. Furthermore, recall in minGRU's derivation we showed that $\log(\sigma(\boldsymbol{k}_t)) = -\text{Softplus}(-\boldsymbol{k}_t)$ Using this, we can simplify the computation as follows:

$$
\begin{aligned}
\log(\boldsymbol{f}'_t) &= -\text{Softplus}\left(\log(\sigma(\boldsymbol{k}_t)) - \log(\sigma(\boldsymbol{p}_t))\right) \\
&= -\text{Softplus}\left(\text{Softplus}(-\boldsymbol{p}_t) - \text{Softplus}(-\boldsymbol{k}_t)\right)
\end{aligned}
$$

Similarly, we also get that:

$$\log(\boldsymbol{i}'_t) = -\text{Softplus}\left(\text{Softplus}(-\boldsymbol{k}_t) - \text{Softplus}(-\boldsymbol{p}_t)\right)$$

Combining these derivations, we get the parallel mode (Algorithm 8) for efficient training.

---

**Algorithm 7** Sequential Mode: Minimal Version of LSTM (minLSTM) with length independence scaling trained in log-space

---

**Input:** $\boldsymbol{x}_t, \boldsymbol{h}_{t-1}$
**Output:** $\boldsymbol{h}_t$
$\quad \boldsymbol{f}_t \leftarrow \sigma(\text{Linear}_{d_h}(\boldsymbol{x}_t))$
$\quad \boldsymbol{i}_t \leftarrow \sigma(\text{Linear}_{d_h}(\boldsymbol{x}_t))$
$\quad \boldsymbol{f}'_t, \boldsymbol{i}'_t \leftarrow \frac{\boldsymbol{f}_t}{\boldsymbol{f}_t + \boldsymbol{i}_t}, \frac{\boldsymbol{i}_t}{\boldsymbol{f}_t + \boldsymbol{i}_t}$
$\quad \tilde{\boldsymbol{h}}_t \leftarrow g(\text{Linear}_{d_h}(\boldsymbol{x}_t))$
$\quad \boldsymbol{h}_t \leftarrow \boldsymbol{f}'_t \odot \boldsymbol{h}_{t-1} + \boldsymbol{i}'_t \odot \tilde{\boldsymbol{h}}_t$

---

**Algorithm 8** Parallel Mode: Minimal Version of LSTM (minLSTM) with length independence scaling for training in log-space

---

**Input:** $\boldsymbol{x}_{1:t}, \boldsymbol{h}_0$
**Output:** $\boldsymbol{h}_{1:t}$
$\quad \text{diff} \leftarrow \text{Softplus}(-\text{Linear}_{d_h}(\boldsymbol{x}_{1:t})) - \text{Softplus}(-\text{Linear}_{d_h}(\boldsymbol{x}_{1:t}))$
$\quad \log\_\boldsymbol{f}'_{1:t} \leftarrow -\text{Softplus}(\text{diff})$
$\quad \log\_\boldsymbol{i}'_{1:t} \leftarrow -\text{Softplus}(-\text{diff})$
$\quad \log\_\boldsymbol{h}_0 \leftarrow \log(\boldsymbol{h}_0)$
$\quad \log\_\tilde{\boldsymbol{h}}_{1:t} \leftarrow \log\_g(\text{Linear}_{d_h}(\boldsymbol{x}_{1:t}))$
$\quad \boldsymbol{h}_{1:t} \leftarrow \text{ParallelScanLog}(\log\_\boldsymbol{f}'_{1:t}, [\log\_\boldsymbol{h}_0, \log\_\boldsymbol{i}'_{1:t} + \log\_\tilde{\boldsymbol{h}}_{1:t})$

---

### B.3 PYTORCH CODE: LOG-SPACE VERSION

```
1 def g(x):
2     return torch.where(x >= 0, x+0.5, torch.sigmoid(x))
3 def log_g(x):
4     return torch.where(x >= 0, (F.relu(x)+0.5).log(),
5                               -F.softplus(-x))
```

Listing 6: The continuous function $g$ ensures that $\tilde{\boldsymbol{h}}_t \leftarrow g(\text{Linear}_{d_h}(x_t))$ is positive.

#### B.3.1 MINGRU: A MINIMAL GRU

```
1     def forward(self, x_t, h_prev):
2         # x_t: (batch_size, 1, input_size)
3         # h_prev: (batch_size, 1, hidden_size)
4
5         z = torch.sigmoid(self.linear_z(x_t))
6         h_tilde = g(self.linear_h(x_t))
7         h_t = (1 - z) * h_prev + z * h_tilde
8         return h_t
```

Listing 7: Sequential Mode: Minimal Version of GRU (minGRU) trained in log-space

```
1     def forward(self, x, h_0):
2         # x: (batch_size, seq_len, input_size)
```

```
 3          # h_0: (batch_size, 1, hidden_size)
 4
 5          k = self.linear_z(x)
 6          log_z = -F.softplus(-k)
 7          log_coeffs = -F.softplus(k)
 8          log_h_0 = log_g(h_0)
 9          log_tilde_h = log_g(self.linear_h(x))
10          h = parallel_scan_log(log_coeffs,
11                  torch.cat([log_h_0, log_z + log_tilde_h], dim=1))
12          return h
```

Listing 8: Parallel Mode: Minimal Version of GRU (minGRU) for training in log-space

### B.3.2 MINLSTM: A MINIMAL LSTM

```
 1      def forward(self, x_t, h_prev):
 2          # x_t: (batch_size, 1, input_size)
 3          # h_prev: (batch_size, 1, hidden_size)
 4
 5          f_t = torch.sigmoid(self.linear_f(x_t))
 6          i_t = torch.sigmoid(self.linear_i(x_t))
 7          tilde_h_t = g(self.linear_h(x_t))
 8          f_prime_t = f_t / (f_t + i_t)
 9          i_prime_t = i_t / (f_t + i_t)
10          h_t = f_prime_t * h_prev + i_prime_t * tilde_h_t
11          return h_t
```

Listing 9: Sequential Mode: Minimal Version of LSTM (minLSTM) with length independence scaling trained in log-space

```
 1      def forward(self, x, h_0):
 2          # x: (batch_size, seq_len, input_size)
 3          # h_0: (batch_size, 1, hidden_size)
 4
 5          diff = F.softplus(-self.linear_f(x))  \
 6                      - F.softplus(-self.linear_i(x))
 7          log_f = -F.softplus(diff)
 8          log_i = -F.softplus(-diff)
 9          log_h_0 = torch.log(h_0)
10          log_tilde_h = log_g(self.linear_h(x))
11          h = parallel_scan_log(log_f,
12                  torch.cat([log_h_0, log_i + log_tilde_h], dim=1))
13          return h
```

Listing 10: Parallel Mode: Minimal Version of LSTM (minLSTM) with length independence scaling for training in log-space

## C    DETAILED EXPERIMENT SETUP

In this section, we describe the experiment setup in detail.

### C.1    DATASETS

**Selective Copying.** In this task, the model learns to extract data tokens from a sequence while disregarding noise tokens. Following Gu & Dao (2024), we consider a vocabulary of 16 and sequences of length 4096. Each sequence includes 16 randomly placed data tokens. The remainder of the tokens are noise.

**Chomsky Hierarchy.** In this task, we consider the Chomsky Hierarchy benchmark (Deletang et al., 2023), which includes a variety of formal language tasks that span different levels of the Chomsky hierarchy. Additionally, we include the two additional tasks described in Beck et al. (2024): Majority and Majority Count. Models are trained on tasks whose sequences vary in length up to 40. Evaluation is conducted for task lengths between 40 and 256 to assess the models' ability to generalize.

**Long Range Arena.** Our experiments on the Long Range Arena benchmark consist of three sequence modelling tasks with sequence lengths from 1024 to 4000, designed to evaluate architectures on long-range modelling:

- **Retrieval:** Based on the ACL Anthology Network (Radev et al., 2009), the task is to classify whether two citations, represented as integer token sequences, are equivalent. Sequences are of length 4000 with two possible classes.

- **ListOps:** An extended version of ListOps (Nangia & Bowman, 2018). The task is to compute the result of a nested mathematical expression in prefix notation. Sequences are of length 2048 with ten possible classes.

- **G-Image:** Based on CIFAR-10 (Krizhevsky, 2009), the task is to predict the class of $32 \times 32$ grayscale images (converted from RGB). Sequences are of length 1024 with ten possible classes.

**Reinforcement Learning.** In this setting, we consider continuous control tasks from the D4RL benchmark (Fu et al., 2020). These tasks based on MuJoCo comprise of three environments with dense rewards: HalfCheetah, Hopper, and Walker. For each environment, three different datasets are considered that have varying level represent varying levels of data quality:

- Medium (M): One million timesteps generated by a policy scoring about one-third of an expert policy's score.

- Medium-Replay (M-R): A replay buffer from an agent trained to perform like the Medium policy.

- Medium-Expert (M-E): One million timesteps from the Medium policy combined with one million from an expert policy.

Following Fu et al. (2020), reported scores are normalized such that 100 represents an expert policy performance.

**Language Modelling.** In this setting, we consider the Shakespeare dataset, comprising a collection of text data derived from the works of William Shakespeare. The training and testing data consists of $1,003,854$ and $111,540$ tokens respectively.

### C.2    ARCHITECTURE

In our work, the primary goal was to demonstrate that simplified RNN architectures, such as minLSTM and minGRU, can perform comparably to modern state-of-the-art sequence models. To achieve this, we stick with a minimalistic architecture, following standard practices such as residual connections, normalization, and a downprojection layer for the RNN's expanded hidden states. For more

complex tasks like language modeling and Long Range Arena, standard components (convolutional layer and MLP) are added[4].

**Selective Copying:** No additional components.

**Chomsky Hierarchy:** (Conv4 → minRNN), i.e., a convolutional layer with a kernel size of 4 is applied temporally before the minimal RNN.

**Long Range Arena:** (Conv4 → minRNN → MLP)

**Language Modelling:** (Conv4 → minRNN → MLP)

**Reinforcement Learning:** (minRNN → MLP)[5]

### C.3 HYPERPARAMETERS AND GENERAL EXPERIMENTAL DETAILS

**Selective Copying.** For this task, we closely follow the setup of Gu & Dao (2024), training the model for 400k steps with a batch size of 64 and an input dimension of 64. Due to GPU memory constraints, gradient accumulation is applied, where gradients for two batches of size 32 are accumulated before each gradient update and clipped to 1.0. The optimizer used is Adam with a learning rate of $3 \times 10^{-4}$ alongside early stopping. Each model consists of 3 layers with a dropout rate of 0.1. The minLSTM and minGRU models have an expansion factor of 6. Baseline results are referenced from the Mamba paper.

**Long Range Arena.** For this benchmark, we closely follow the setup of Beck et al. (2024). For Retrieval, the models consisted of 6 blocks and an embedding dimension of 128 and were trained with a batch size of 64. For ListOps, the models consisted of 8 blocks and an embedding dimension of 128 and were trained with a batch size of 32. For G-Image, the models consisted of 6 blocks and an embedding dimension of 512 and were trained with a batch size of 64. All models were trained for 250k steps using AdamW optimizer with a learning rate of 0.001, weight decay of 0.05, 10% linear warm-up steps, and cosine annealing.

**Chomsky Hierarchy.** For this benchmark, we closely follow the setup of Beck et al. (2024), training models consisting of two blocks. The models were trained for 500k steps with a batch size of 256 and the AdamW optimizer with a learning rate of $3 \times 10^{-4}$ and weight decay of 0.01.

**Language Modelling.** The models are optimized using AdamW with a learning rate of $1 \times 10^{-3}$. Each model consists of three layers, a dropout ratio of 0.2, and an embedding dimension of 384. Training is done with 5k steps using a batch size of 64 and evaluated every 25 steps. Gradients are clipped to 0.25. The Transformer is configured with 6 heads. Mamba uses an SSM state expansion factor of 16 and a block expansion factor of 2. Following Mamba, both minLSTM and minGRU utilize an expansion factor of 2 as well.

**Reinforcement Learning.** We follow the hyperparameter settings outlined by Ota (2024). For Hopper (Medium) and Hopper (Medium-Replay), an embedding dimension of 256 is used, while all other environments utilize an embedding dimension of 128. The learning rate is set to $1 \times 10^{-4}$ for Hopper (Medium), Hopper (Medium-Replay), and Walker (Medium). For all other environments and datasets, the learning rate is $1 \times 10^{-3}$. The models are optimized using AdamW with a weight decay of $1 \times 10^{-4}$ and a linear warmup for $10,000$ steps. Each model consists of 3 layers and has a dropout ratio of 0.1. The models are trained for 100k steps with a batch size of 64. Results for the baselines are referenced from the Mamba and Aaren papers.

---

[4]There is a trend in modern recurrent sequence models of prepending a convolutional layer (kernel size of 4) before their recurrent unit – for example, see Mamba (Gu & Dao, 2024) and xLSTM (Beck et al., 2024) Empirically, we found that including this convolutional layer also helped minRNNs.

[5]Note this is equivalent to the standard Decision Transformer framework for (Offline) RL, replacing the self-attention module with minLSTM or minGRU.

## D    ADDITIONAL EXPERIMENTS

### D.1    CHOMSKY HIERARCHY + LONG RANGE ARENA

In this section, we conduct experiments on both the Chomsky Hierarchy (Deletang et al., 2023) and Long Range Arena (Tay et al., 2021) benchmarks, which are well-established in the literature for evaluating sequence models. Together, these benchmarks provide a test of a model's ability to generalize and handle long-range dependencies, which are crucial for modern sequence modelling tasks.

We compare Minimal RNNs against other fully parallelizable models, such as RWKV, Mamba, and xLSTM[1:0] (using its parallelizable mLSTM module). Following Beck et al. (2024), we focus on tasks from the Chomsky Hierarchy where models have achieved at least 30% accuracy, indicating partial solvability. We closely followed the hyperparameter configurations outlined in the xLSTM paper and averaged results over 3 seeds for consistency. The baseline results (accuracy – higher is better) are taken from the xLSTM paper (Figure 4 for Chomsky Hierarchy and Table 6 for Long Range Arena).

Our experiments (Table 4 and extended Table 5) show that Minimal RNNs achieve competitive performance with state-of-the-art models (e.g., Mamba and xLSTM) across all tasks on these benchmarks, outperforming other models such as Retention, Hyena, RWKV, and Llama.

### D.2    INFERENCE RUNTIME COMPARISON

In these experiments, we compare the inference speeds of GRU, LSTM, minGRU, minLSTM, and Mamba (using the official implementation). It is important to note that inference speed may vary depending on the hardware and implementation used.

For this analysis, we tested different batch sizes (8, 16, 32, 64) and sequence lengths (up to 2048). In Figure 3, we present the average inference speed across 50 runs, taking context tokens into account before performing inference. Since GRU and LSTM models process context tokens sequentially, their inference times are considerably slower than those of minGRU, minLSTM, and Mamba, all of which benefit from parallel processing.

Overall, minLSTM and minGRU show inference speeds comparable to Mamba. Specifically, minGRU was 6.6%, 4.1%, 4.9%, and 2.9% faster than Mamba for batch sizes of 8, 16, 32, and 64, respectively. On the other hand, minLSTM was 3.6%, 2.9%, 0%, and 1.3% slower than Mamba for the same batch sizes.

Since minLSTM and minGRU are simplifications of LSTM and GRU, we expect them to generally perform faster, including during inference. This is demonstrated in Figure 4, where we compare the inference speed of minLSTM and minGRU with traditional LSTM and GRU models across varying batch sizes. As expected, minGRU and minLSTM are 19.6% and 41.5% faster than GRU and LSTM for a batch size of 64, respectively.

### D.3    ARCHITECTURE ABLATION

In our work, the main objective was to demonstrate that simplified RNN architectures, such as minLSTM and minGRU, can perform on par with modern state-of-the-art sequence models. To achieve this, we adopted a minimalistic architectural design, incorporating standard practices such as residual connections, normalization, and a downprojection layer for the RNN's expanded hidden states. For more complex tasks, like language modeling and Long Range Arena, we introduced a convolutional layer and a multi-layer perceptron (MLP).

To better understand the impact of these architectural choices, we conducted an ablation study on the ListOps (Long Range Arena) dataset of these additional components. The results, averaged over 3 seeds, are shown in Table 6. The table highlights the effect of adding different layers to the minLSTM model. For ListOps, incorporating a convolutional layer (Conv) and an MLP resulted in improved performance.

| Method | | Bucket Sort | Missing Duplicate | Cycle Nav. | Even Pairs | Majority |
|---|---|---|---|---|---|---|
| Transformers | Llama | 0.92 | 0.08 | 0.04 | 1.0 | 0.37 |
| Modern RNNs | Mamba | 0.69 | 0.15 | 0.86 | 1.0 | 0.69 |
| | RWKV-4 | 0.54 | 0.21 | 0.13 | 1.0 | 0.63 |
| | xLSTM | 0.97 | 0.33 | 0.86 | 1.0 | 0.74 |
| Minimal RNNs | minLSTM (Ours) | 0.94 | 0.26 | 0.79 | 1.0 | 0.93 |

| Method | | Majority Count | Retrieval | ListOps | G-Image | Average |
|---|---|---|---|---|---|---|
| Transformers | Llama | 0.13 | 0.85 | 0.38 | 0.54 | 0.48 |
| Modern RNNs | Mamba | 0.45 | 0.90 | 0.33 | 0.69 | 0.64 |
| | RWKV-4 | 0.13 | 0.90 | 0.39 | 0.69 | 0.51 |
| | xLSTM | 0.46 | 0.91 | 0.41 | 0.70 | 0.71 |
| Minimal RNNs | minLSTM (Ours) | 0.47 | 0.89 | 0.59 | 0.67 | 0.73 |

Table 4: **Results for Chomsky Hierarchy and Long Range Arena Benchmarks.** We compare minLSTM against other fully parallelizable models, including RWKV, Mamba, and xLSTM[1:0] (using the mLSTM module). The baseline results (accuracy – higher is better) are taken from the xLSTM paper (Figure 4 for Chomsky Hierarchy and Table 6 for Long Range Arena). The results demonstrate that minLSTM achieves competitive performance with state-of-the-art models such as Mamba and xLSTM across all tasks on these benchmarks.

### D.4  INITIALIZATION ANALYSES

In this set of experiments, we examine the effect of initialization on the model's performance. Depending on the task at hand, it may be beneficial to encourage the model to retain information over time. One way to achieve this is by increasing the bias term in the forget gate of the minLSTM, which promotes information retention earlier in the training process. As a result, the forget gate $f_t$ of the LSTM approaches a value of 1 due to this new initialization. As shown in Figure 5, increasing the forget gate bias in minLSTM enhances training efficiency, leading to faster convergence and greater stability during training.

## E  ADDITIONAL RELATED WORK

**Parallel Scan.** Generalizing across the families of methods (including minLSTM and minGRU), these recent sequence models can be viewed as members of the same family of functions trainable via a parallel scan: $v_t = a_t \odot v_{t-1} + b_t$ (see Section 2.3) where $a_t$ and $b_t$ are functions of the input token $x_t$. Improving upon the parallel scan algorithm, several models (Yang et al., 2024; Gu & Dao, 2024) such as Mamba have proposed specialized hardware-efficient methods that leverage GPU's memory hierarchy to reduce high I/O costs and speed up training. In our work, we implemented minLSTM and minGRU in plain PyTorch. However, due to the structural similarities in recurrences amongst the numerous methods that leverage parallel scan, many techniques such as chunking that apply to one work for speeding up training can also apply to others such as minGRU and minLSTM.

**Parameter Initializations.** Unrolling the recurrences of these new recurrent sequence models over time often results in their outputs and gradients vanishing/exploding (Wang et al., 2024b) due to time dependency in their output's scale. To ensure model stability, the parameters of many models such as state-space models are initialized according to special distributions (Gu et al., 2020; 2022; Orvieto et al., 2023). In contrast, we found that minLSTM and minGRU are already stable using the default PyTorch initialization. Unlike SSMs, minLSTM and minGRU's outputs are time-independent in scale, avoiding potential instabilities.

| Method | | Context Sensitive | | Regular | | xLSTM | |
|--------|--|-------------------|--|---------|--|-------|--|
| | | Bucket Sort | Missing Duplicate | Cycle Nav. | Even Pairs | Majority | Majority Count |
| Traditional RNNs | GRU | 0.54 | 0.29 | 0.94 | 1.0 | 0.60 | 0.42 |
| | LSTM | 0.99 | 0.33 | 1.0 | 1.0 | 0.54 | 0.46 |
| Transformers | Llama | 0.92 | 0.08 | 0.04 | 1.0 | 0.37 | 0.13 |
| Modern RNNs | Mamba | 0.69 | 0.15 | 0.86 | 1.0 | 0.69 | 0.45 |
| | Retention | 0.13 | 0.03 | 0.05 | 0.51 | 0.36 | 0.12 |
| | Hyena | 0.3 | 0.06 | 0.06 | 0.93 | 0.36 | 0.18 |
| | RWKV-4 | 0.54 | 0.21 | 0.13 | 1.0 | 0.63 | 0.13 |
| | RWKV-5 | 0.49 | 0.15 | 0.26 | 1.0 | 0.73 | 0.34 |
| | RWKV-6 | 0.96 | 0.23 | 0.31 | 1.0 | 0.76 | 0.24 |
| | xLSTM | 0.97 | 0.33 | 0.86 | 1.0 | 0.74 | 0.46 |
| Minimal RNNs | minLSTM (Ours) | 0.94 | 0.26 | 0.79 | 1.0 | 0.93 | 0.47 |

Table 5: **Extended Results for Chomsky Hierarchy Benchmark.** The baseline results (accuracy — higher is better) are taken from the xLSTM paper (Figure 4). We compare minLSTM against other fully parallelizable models, including RWKV, Mamba, and xLSTM[1:0] (using the mLSTM module). The results demonstrate that minLSTM achieves competitive performance with state-of-the-art models such as Mamba and xLSTM, while outperforming other models like Retention, Hyena, RWKV, and Llama across all tasks in the Chomsky Hierarchy benchmark.

| Model | Accuracy |
|-------|----------|
| minLSTM | 0.46 |
| minLSTM (+ Conv) | 0.45 |
| minLSTM (+ MLP) | 0.52 |
| minLSTM (+ Conv + MLP) | 0.59 |

Table 6: **Architecture Ablation on the ListOps (Long Range Arena) Dataset.** Results (accuracy – higher is better) are averaged over 3 seeds. The table shows the impact of adding different layers to the minLSTM model. For more complex tasks like language modeling and Long Range Arena, we incorporate a convolutional layer (Conv) and a multi-layer perceptron (MLP). The performance increases when these components are added.

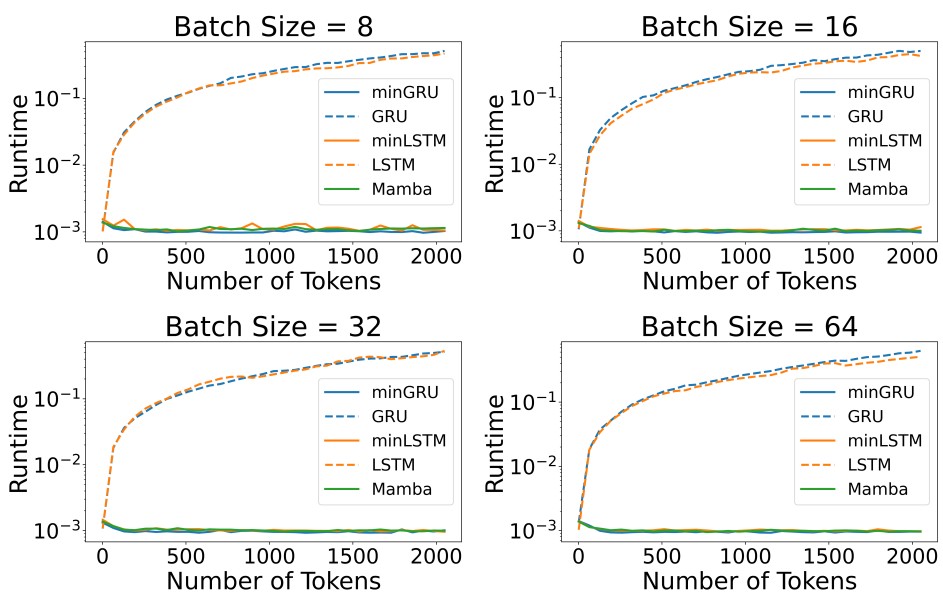

Figure 3: **Runtime Comparison of Inference with Context Tokens: Parallelizable RNNs (minL-STM, minGRU, and Mamba) vs. Traditional RNNs (LSTM and GRU).** As sequential models, LSTM and GRU exhibit significantly slower inference times when processing an increasing number of context tokens, compared to the parallelizable models minLSTM, minGRU, and Mamba.

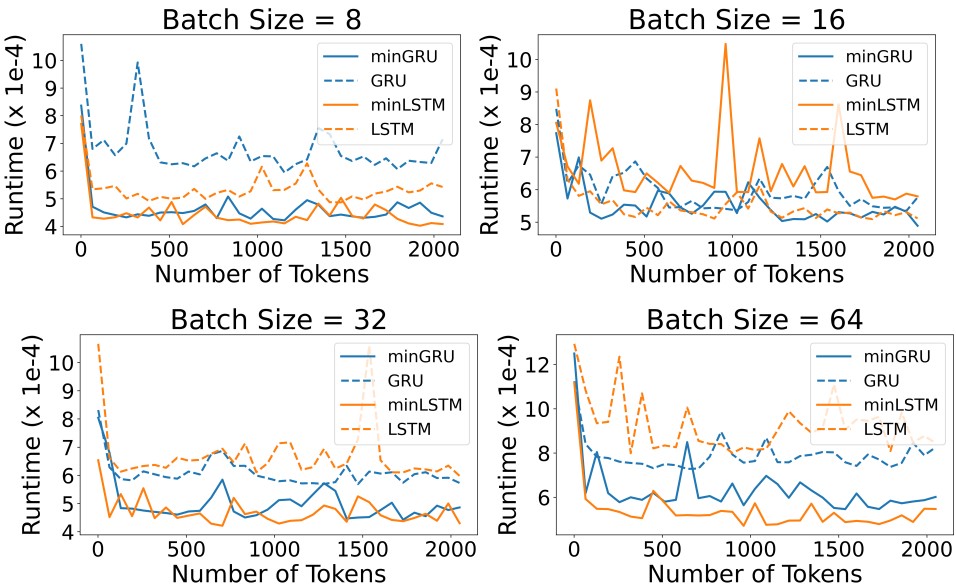

Figure 4: **Runtime Comparison of Inference: Minimal RNNs (minLSTM and minGRU) vs. Traditional Counterparts (LSTM and GRU).** As simplified versions of LSTM and GRU, minL-STM and minGRU generally exhibit faster inference times, particularly with larger batch sizes, as shown in the plots.

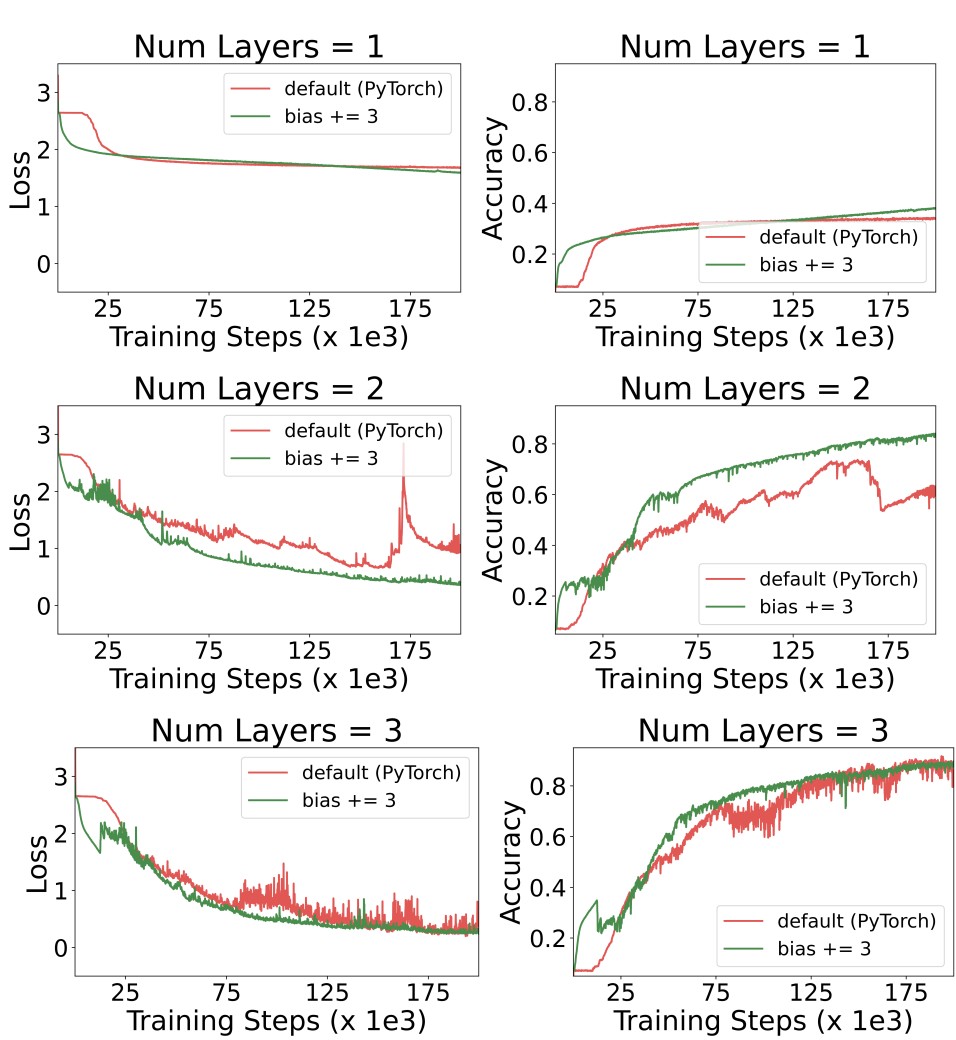

Figure 5: **Impact of Forget Gate Bias Initialization on Training Efficiency.** The plot illustrates how increasing the bias of the forget gate in minLSTM enhances training efficiency by promoting earlier retention of information, leading to faster convergence and a more stable training process.

