# OpenReview forum: "Were RNNs All We Needed?"
_ICLR.cc/2025/Conference — Submitted to ICLR 2025_

### Official Review · Reviewer_rQhu · 2024-10-15

**Soundness:** 4
**Presentation:** 4
**Contribution:** 1
**Rating:** 3
**Confidence:** 4

**Summary:**

Transformers are expensive to run on long sequences due to their quadratic complexity. Traditional RNNs such as LSTMs and GRUs have cheaper inference costs, but they cannot be parallelized during training. This paper shows how we can minimally modify these architectures to train them with parallel scans. This greatly reduces training runtime, while still maintaining good empirical performance (as demonstrated on small scale reinforcement learning and language modelling tasks).

**Strengths:**

The paper is well written and easy to follow. It convincingly demonstrates the speed benefits of min LSTM and GRU over their sequential counterparts.

**Weaknesses:**

I am afraid that the paper is half a year to one year late. Both the xLSTM (Beck et al. 2024) and the MatMul-free (Zhu et al. 2024) papers leverage a similar insight to the one of this paper, that is removing everything in LSTM/GRUs that cannot be parallelized during training. Those papers demonstrate the effectiveness of this approach at a larger scale (more than 10x the number of parameters). In addition to that, detailed comparison to these works is missing. For these reasons, I think the paper requires heavy modifications (e.g. more detailed experiments, at a larger scale) before being accepted.

**Questions:**

If my arguments above are valid, I don't have any question that can change my opinion of the paper.

---

> ### Author Response · Authors · 2024-11-21
> **Response to Reviewer rQhu (1/2)**
>
> We would like to thank the reviewer for their comments and feedback.
>
> > I am afraid that the paper is half a year to one year late. Both the xLSTM (Beck et al. 2024) and the MatMul-free (Zhu et al. 2024) papers leverage a similar insight to the one of this paper, that is removing everything in LSTM/GRUs that cannot be parallelized during training. Those papers demonstrate the effectiveness of this approach at a larger scale (more than 10x the number of parameters). In addition to that, detailed comparison to these works is missing. For these reasons, I think the paper requires heavy modifications (e.g. more detailed experiments, at a larger scale) before being accepted.
>
>
> Thank you for your feedback. We would like to note that the ICLR deadline was on October 1st, 2024. Both xLSTM (Beck et al., 2024) and MatMul-free (Zhu et al., 2024) were recently released as preprints on arXiv on May 7th, 2024 and June 4th, 2024, respectively.
>
> According to ICLR’s standards, a paper is considered contemporaneous if it was posted within four months prior to the submission deadline. Since MatMul-free was released within this four-month window, it is contemporaneous with our submission. xLSTM, on the other hand, falls just outside this window. To address your concerns, we have provided new experiments and comparisons with xLSTM.
>
>
> ## Conceptual Difference compared to xLSTM and MatMul-free
>
> Firstly, however, we want to emphasize that our approach is conceptually different from both xLSTM and MatMul-free.
>  - xLSTM focuses on extending and scaling LSTM architectures, incorporating more components such as exponential gates and additional normalization states. More concretely, xLSTM is a hybrid model composed of two variants: sLSTM and mLSTM. sLSTM retains the hidden state, making it sequential-only and less efficient. mLSTM removes the hidden state dependencies to enable parallelization, introduces a matrix memory cell, and uses a query vector for retrieval from the memory. This makes mLSTM more efficient and parallelizable but still retains considerable complexity.
>  - MatMul-free, on the other hand, removes matrix multiplications from both dense layers and self-attention mechanisms by replacing them with simpler operations, such as element-wise Hadamard products. It also uses ternary weight quantization and proposes hardware optimizations (GPU and FPGA implementations) to improve efficiency.
>
> In contrast, our approach simplifies the traditional GRU and LSTM architectures by removing components that cannot be parallelized during training (hidden state dependencies) and range restrictions (tanh activation). This results in a fully parallelizable and computationally efficient architecture that remains simple and highly implementable, without the added complexity of hybrid models or hardware-specific optimizations.
>
> Our main contribution is to show that these simplified versions of decade-old RNN architectures can achieve competitive performance across a range of benchmarks, challenging the prevailing trend in the community toward increasing architectural and algorithmic complexity, as seen in models like Mamba, xLSTM, and S4.

---

> ### Author Response · Authors · 2024-11-21
> **Response to Reviewer rQhu (2/2)**
>
> ## Comparison with xLSTM
>
> To address your concern regarding the comparison to recent works, we have expanded our experimental evaluation to include two well-established benchmarks for sequence modeling used in the xLSTM paper: Long Range Arena (Tay et al., 2021) and Chomsky Hierarchy (Delétang et al., 2023).
> We compare Minimal RNNs against other fully parallelizable models, such as RWKV, Mamba, and xLSTM[1:0] (using the mLSTM module). We closely followed the hyperparameter configurations outlined in the xLSTM paper and averaged results over 3 seeds for consistency. The baseline results (accuracy – higher is better) are retrieved from the xLSTM paper (Figure 4 for Chomsky Hierarchy and Table 6 for Long Range Arena).
>
> Empirically, we find that Minimal RNNs achieve comparable performance to these state-of-the-art models on challenging tasks. This demonstrates that simplified decades-old RNNs can compete effectively with more complex modern sequence models. We have included the extended table with these results in the Appendix for reference.
>
> |                     | Bucket Sort | Missing Duplicate | Cycle Nav. | Even Pairs | Majority | Majority Count | Retrieval | ListOps | G-Image | Average |
> |:-------------------:|:-----------:|:-----------------:|:----------:|:----------:|:--------:|:--------------:|:---------:|:-------:|:-------:|:-------:|
> | Llama (Transformer) |     0.92    |        0.08       |    0.04    |     1.0    |   0.37   |      0.13      |    0.85   |   0.38  |   0.54  |   0.48  |
> |        Mamba        |     0.69    |        0.15       |    0.86    |     1.0    |   0.69   |      0.45      |    0.90   |   0.33  |   0.69  |   0.64  |
> |        RWKV-4       |     0.54    |        0.21       |    0.13    |     1.0    |   0.63   |      0.13      |    0.90   |   0.39  |   0.69  |   0.51  |
> |        xLSTM        |     0.97    |        0.33       |    0.86    |     1.0    |   0.74   |      0.46      |    0.91   |   0.41  |   0.70  |   0.71  |
> |    minLSTM (Ours)   |     0.94    |        0.26       |    0.79    |     1.0    |   0.93   |      0.47      |    0.89   |   0.59  |   0.67  |   0.73  |
>
> While our current scaling is constrained by hardware limitations (16 GB GPUs), we believe these results on popular benchmarks in conjunction with our training resource analyses (Figure 1) clearly demonstrate that Minimal RNNs can handle long-range dependencies and scale efficiently similar to more complex models.
>
>
> ---
>
> Beck, Maximilian, et al. "xLSTM: Extended Long Short-Term Memory." arXiv preprint 	arXiv:2405.04517 (2024).
>
> Zhu, Rui-Jie, et al. "Scalable MatMul-free Language Modeling." arXiv preprint arXiv:2406.02528 (2024).
>
> Tay, Yi, et al. "Long Range Arena: A Benchmark for Efficient Transformers." ICLR (2021).
>
> Delétang, Grégoire, et al. "Neural Networks and the Chomsky Hierarchy." ICLR (2023).

---

> > ### Comment · Reviewer_rQhu · 2024-11-21
> >
> > I acknowledge the author's answers to my review as well as their answer to the other reviewers.
> >
> > I appreciate the author's efforts in providing a comprehensive comparison of the minLSTM with existing methods. It definitely strengthens the empirical claims of the paper.
> >
> > **Re the comparison with xLSTM and matmul free GRU.** While I agree that the proposed method differs from these architectures, I believe that introducing these modifications is mostly needed when scaling the networks to larger language modeling tasks. I understand that these experiments require large infrastructures than what most researchers have access to, but I do think that testing these ideas must be done at a reasonably large scale (e.g. 500m to 1B for language modelling), or on different modalities (e.g. RL/DNA), in order to provide the community with solid insights.
> >
> > **Re novelty**. The papers I mention were indeed published within a 4 month window before ICLR deadline. I find it hard to estimate what would have been my review without knowing about these two papers. On one hand, the paper provides nice insights and comprehensive experiments on relatively small tasks (the rebuttal definitely improved that). On the other hand, it lacks the scale at which we can differentiate architectures and the proposed architecture is not specialized enough to shine on very specific tasks (e.g. LRA, like S4 would). Taking that into account, my score would have probably been 6.
> > 4 months is almost an eternity in the field of designing alternative architectures to Transformers, so I keep my score as it is (3). That said, I let the AC decide which score (3 or 6) they consider the most applicable to the current situation.

---

> ### Author Response · Authors · 2024-11-22
>
> We would like to thank the reviewer for their thoughtful response and constructive feedback.
>
> > I do think that testing these ideas must be done at a reasonably large scale (e.g. 500m to 1B for language modelling), or on different modalities (e.g. RL/DNA), in order to provide the community with solid insights.
>
> We fully acknowledge the importance of testing these models at larger scales and across different modalities (e.g., RL/DNA) to assess their real-world applicability more effectively. We would like to point out that we have included experiments on reinforcement learning (RL) and demonstrated competitive performance against recent RL methods on a range of locomotion tasks (Table 3).
>
> Lastly, we would also like to highlight that Minimal RNNs are easily implementable in a few lines of code (e.g., Listing 8). Considering Minimal RNNs ease of implementation and promising properties (in terms of computational resources needed and their competitive performance), we believe the paper would be of interest to the ICLR and broader machine learning community.

---

> > ### Public Comment · ~Rui-Jie_Zhu1 · 2024-11-27
> > **Comment from the developer of Matmul-free LM**
> >
> > I noticed this interesting work and the reviewer's discussion of MatMul-free, and I would like to offer some clarifications as the first author of MatMul-free.
> >
> > I appreciate how this paper effectively demonstrates approaches to parallelize GRU/LSTM architectures and presents these concepts in an accessible manner. Regarding the authors' characterization of MatMul-free, I would like to provide some additional context.
> >
> > While it's correct that MatMul-free removes matrix multiplications and utilizes element-wise operations, this represents only part of our contribution. A key aspect of our work was also revisiting and modernizing GRU as a parallelizable token mixer, building upon insights from HGRN. The minGRU approach seems to follow a similar path to HGRN, which itself drew inspiration from earlier work on efficient RNN variants using element-wise linear recurrence relations, as demonstrated by GILR.
> > Specifically, we revisited GRU architecture and adapted it into a modern, parallelizable token mixer, acknowledging in our paper that this design drew from HGRN. We demonstrated that such a token mixer, using only element-wise products, avoided introducing matrix multiplications compared to other 2D hidden state Linear Transformers.
> >
> > The clear presentation makes a valuable contribution to the community, but I also align with [Songlin's earlier comment](https://openreview.net/forum?id=GrmFFxGnOR&noteId=2QR0ZJjvCm) that a more thorough discussion of prior work (particularly HGRN and GILR) would further strengthen the paper. In our own work, we made sure to acknowledge HGRN's contributions, and HGRN in turn recognized GILR as an important early effort in this direction. It's worth noting that our MatMul-free work represents just one branch in the broader exploration of linear recurrence, focusing specifically on achieving matmul-free token mixing through linear recurrence.
> >
> > Thanks,
> >
> > Rui-Jie

---

### Official Review · Reviewer_jHLY · 2024-10-29

**Soundness:** 3
**Presentation:** 3
**Contribution:** 1
**Rating:** 3
**Confidence:** 4

**Summary:**

The paper introduces minimal variants of LSTM and GRU (minLSTM and minGRU) that enable parallel training via the parallel scan algorithm. Key modifications include removing hidden state dependencies and rescaling mechanisms in the LSTM. The authors demonstrate competitive performance with modern architectures on several tasks.

**Strengths:**

The paper's technical exposition is clear, offering good motivation for removing parts of the RNN architectures. The authors provide thorough motivation for each architectural decision, especially regarding the time-independence properties of their models. The connection to parallel scan algorithms makes sense.

The architectural innovations demonstrate consideration of practical implementation challenges. The rescaling mechanism in minLSTM ensures time-independence while maintaining model expressivity. The authors have given careful thought to numerical stability issues, providing both standard and log-space formulations of their approach. It is nice to see the PyTorch implementation details, which help to make the algorithm very concrete.

The experimental validation covers a range of tasks and provides evidence of training speed advantages wrt traditional RNNs. The performance on the selective copying task is particularly impressive, demonstrating the model's ability to handle a particular type of long-range dependencies.

**Weaknesses:**

The most significant concern is the paper's relationship to prior work, specifically Martin & Cundy (ParalleIizing Linear Recurrent Neural Nets over Sequence Length, ICLR 2018), which appears to have developed very similar ideas. The proposed minGRU architecture appears mathematically equivalent to their GILR architecture when \(g_t = 1 - z_t\) and \(i_t = \tilde{h}_t\) in the authors' notation. The parallel scan approach for training is also very similar. While the LSTM rescaling mechanism appears novel, this substantial overlap significantly reduces the paper's claimed novelty.

The experimental evaluation, while broad, has several limitations. The benefits over baselines are modest in most tasks, with the exception of the selective copying task, which seems carefully chosen to showcase the method's strengths. The language modeling evaluation relies solely on the Shakespeare dataset, which is both too small by modern standards. This choice of dataset is not representative of real-world language modeling challenges, which typically do not have overfitting as a major worry.

The methodology would benefit from more thorough analysis in several areas. The authors could strengthen their contribution by providing more extensive ablation studies on their architectural choices. The paper would also benefit from a clearer discussion of scaling behavior, to indicate if this architecture could be a potential replacement for transformers or Mamba at the larger scales of modern models.

**Questions:**

Can the authors provide a detailed comparison with Martin & Cundy (2018), specifically addressing how minGRU differs from GILR, if at all, and what novel contributions this work makes beyond the prior work? Additionally, it would be valuable to understand why these ideas are being revisited now and what new insights are gained.

How do the proposed new architectures fare at larger data sizes? How do these models perform on standard language modeling benchmarks? Results on datasets with training sequences longer than 1000 steps would be particularly good to see. Understanding the scaling behavior on modern language modeling tasks is crucial for assessing the practical value of this approach.

Finally, more detailed inference speed comparisons would strengthen the paper. How do minGRU and minLSTM compare with Mamba and traditional LSTM/GRU implementations across different sequence lengths and batch sizes?

---

> ### Author Response · Authors · 2024-11-21
> **Response to Reviewer jHLY (1/4)**
>
> We would like to thank the reviewer for their comments and helpful feedback.
>
> > The most significant concern is the paper's relationship to prior work, specifically Martin & Cundy (ParalleIizing Linear Recurrent Neural Nets over Sequence Length, ICLR 2018), which appears to have developed very similar ideas. The proposed minGRU architecture appears mathematically equivalent to their GILR architecture when (g_t = 1 - z_t) and (i_t = \tilde{h}_t) in the authors' notation. The parallel scan approach for training is also very similar. While the LSTM rescaling mechanism appears novel, this substantial overlap significantly reduces the paper's claimed novelty. [...]
> >
> > Can the authors provide a detailed comparison with Martin & Cundy (2018), specifically addressing how minGRU differs from GILR, if at all, and what novel contributions this work makes beyond the prior work?
> > Additionally, it would be valuable to understand why these ideas are being revisited now and what new insights are gained.
>
>
> Thank you for pointing out the similarities to the work by Martin & Cundy (2018). We would like to clarify the conceptual distinctions between our work and theirs, as well as highlight the novel contributions of our approach.
>
> **Comparison to Martin & Cundy (2018):**
>  - Martin & Cundy (2018) focus on parallelizing linear RNNs and propose the GILR (Generalized Linear RNN) architecture. GILR is used as a linear surrogate for the hidden state dependencies of traditional LSTMs, allowing for parallelization. The resulting architecture GILR-LSTM retains much of the complexity of LSTMs but with parallelizability, resulting in a larger memory footprint due to the use of surrogate states.
>  - In contrast, our work takes a different approach by simplifying traditional RNN architectures rather than augmenting them. We propose to remove components of GRUs and LSTMs, such as the hidden state dependencies and the non-linearities (tanh), to achieve a minimal architecture that is both fully parallelizable and computationally more efficient.
>
> **Novel Contribution:**
>  - While Martin & Cundy (2018) propose a more complex modification to LSTMs with surrogate linear models, we show that simplified, minimal versions of traditional RNNs (such as minGRU and minLSTM) can match the performance of modern, state-of-the-art sequence models (2018 - Present). This is particularly notable given the increasing architectural and algorithmic complexity of many modern sequence models, such as Mamba, xLSTM, and S4. This challenges the prevailing trend in recent literature that increasing model complexity (e.g., through state-space models or attention mechanisms) is necessary for achieving competitive performance.
>
> **Why Revisit These Ideas?:**
>  - Due to the scalability limitations of Transformers, there has been a resurgence of interest in recurrent sequence models, driven by innovations in state-space models (e.g., Mamba -- 1300+ citations, S4 - 1200+ citations) and linear attention mechanisms (e.g., Linear Attention - 1500+ citations, RWKV - 350+ citations). Many recent models require significant architectural and algorithmic complexity to achieve competitive performance on modern tasks.
>  - Our work provides a counterpoint to this trend by showing that simplified RNNs perform competitively with these recent state-of-the-art methods. This provides new insights into the trade-offs between simplicity and complexity in sequence modeling, suggesting that decades-old models may not necessarily be outperformed by modern complex ones.
>
> We have updated the manuscript to clarify these distinctions, particularly in relation to Martin & Cundy’s work, and to highlight our novel contributions in the context of the recent surge in complex sequence modelling approaches.

---

> > ### Author Response · Authors · 2024-11-21
> > **Response to Reviewer jHLY (2/4)**
> >
> > > The experimental evaluation, while broad, has several limitations. The benefits over baselines are modest in most tasks, with the exception of the selective copying task, which seems carefully chosen to showcase the method's strengths.
> >
> > We would like to clarify that the primary goal of our work is not to achieve the best performance on individual tasks, but to demonstrate that simplifying traditional, decades-old RNN architectures can yield results comparable to modern, state-of-the-art sequence models.
> >
> > We selected the Selective Copying task because it was used in the Mamba paper (Gu & Dao, 2023) as one of the primary benchmarks to highlight how much their state-space model, S6, and the Mamba module improved over prior state-of-the-art models such as S4 and Hyena. Mamba is a widely recognized paper that builds on earlier work in state-space models and has garnered over 1300 citations in less than a year.
> >
> > In our work, we show that simplified versions of decades-old RNNs can achieve comparable performance to Mamba on this challenging task. This result is both surprising and valuable, as it challenges the prevailing assumption in the sequence modelling community that increasing architectural and algorithmic complexity is necessary to achieve state-of-the-art performance.

---

> ### Author Response · Authors · 2024-11-21
> **Response to Reviewer jHLY (3/4)**
>
> > The language modeling evaluation relies solely on the Shakespeare dataset, which is both too small by modern standards. This choice of dataset is not representative of real-world language modeling challenges, which typically do not have overfitting as a major worry. [...]
> >
> > How do the proposed new architectures fare at larger data sizes? How do these models perform on standard language modeling benchmarks? Results on datasets with training sequences longer than 1000 steps would be particularly good to see. Understanding the scaling behavior on modern language modeling tasks is crucial for assessing the practical value of this approach. [...]
> >
> > The paper would also benefit from a clearer discussion of scaling behavior, to indicate if this architecture could be a potential replacement for transformers or Mamba at the larger scales of modern models. [...]
>
>
> Thank you for your insightful feedback regarding the scaling behavior and dataset choice in our evaluation. We acknowledge that the Shakespeare dataset, while commonly used for smaller-scale tasks, may not fully capture the complexities of real-world language modeling, particularly in terms of overfitting. Our primary goal in using this dataset was to demonstrate the feasibility of simplified RNN architectures in a controlled environment. However, we recognize the importance of evaluating these models on larger and more challenging datasets to better assess their practical applicability.
>
> In response to your concerns, we have expanded our evaluation to include the Long Range Arena (Tay et al., 2021) and Chomsky Hierarchy benchmarks (Delétang et al., 2024), which include tasks involving longer sequences (all tasks in Long Range Arena exceed 1000 steps). Together, these benchmarks are designed to test a model's ability to generalize across real-world sequence modeling challenges and handle long range dependencies.
>
> For these tasks, we closely follow the hyperparameters of xLSTM (Beck et al., 2024), including model size, embedding dimensions, and number of layers. The baseline results (accuracy – higher is better) are retrieved from the xLSTM paper (Figure 4 for Chomsky Hierarchy and Table 6 for Long Range Arena). We compare against other parallelizable modern sequence models such as Mamba and xLSTM[1:0].
> Our results demonstrate that Minimal RNNs perform competitively with state-of-the-art models like Mamba and xLSTM across these benchmarks. An extended table with additional results and detailed discussions is included in the paper.
>
> |                     | Bucket Sort | Missing Duplicate | Cycle Nav. | Even Pairs | Majority | Majority Count | Retrieval | ListOps | G-Image | Average |
> |:-------------------:|:-----------:|:-----------------:|:----------:|:----------:|:--------:|:--------------:|:---------:|:-------:|:-------:|:-------:|
> | Llama (Transformer) |     0.92    |        0.08       |    0.04    |     1.0    |   0.37   |      0.13      |    0.85   |   0.38  |   0.54  |   0.48  |
> |        Mamba        |     0.69    |        0.15       |    0.86    |     1.0    |   0.69   |      0.45      |    0.90   |   0.33  |   0.69  |   0.64  |
> |        RWKV-4       |     0.54    |        0.21       |    0.13    |     1.0    |   0.63   |      0.13      |    0.90   |   0.39  |   0.69  |   0.51  |
> |        xLSTM        |     0.97    |        0.33       |    0.86    |     1.0    |   0.74   |      0.46      |    0.91   |   0.41  |   0.70  |   0.71  |
> |    minLSTM (Ours)   |     0.94    |        0.26       |    0.79    |     1.0    |   0.93   |      0.47      |    0.89   |   0.59  |   0.67  |   0.73  |
>
> It is important to note that while models like Mamba and xLSTM were run on modern A100 GPUs with 80 GB of memory, our experiments were conducted on older GPUs (P100, T4, and Quadro 5000) with only 16 GB of memory (roughly 20% of the memory available to the other models). This significant memory limitation restricted our ability to test on large-scale language modeling datasets.
>
> Nonetheless, our results on these popular benchmarks are promising—particularly for tasks involving sequences exceeding 1000 steps. These experiments, together with the memory efficiency analysis in Figure 1 showing that Minimal RNNs are more efficient than Mamba, demonstrate that Minimal RNNs can effectively handle long-range dependencies while exhibiting promising scaling behavior and efficiency. This highlights their potential for real-world sequence modeling applications. We have included these new experiments and analyses in the Appendix, as well as clarifications regarding hardware limitations, in the Limitations section.

---

> ### Author Response · Authors · 2024-11-21
> **Response to Reviewer jHLY (4/4)**
>
> > The authors could strengthen their contribution by providing more extensive ablation studies on their architectural choices.
>
> In our work, the primary goal was to demonstrate that simplified RNN architectures, such as minLSTM and minGRU, can perform comparably to modern state-of-the-art sequence models. To achieve this, we intentionally designed a minimalistic architecture, following standard practices such as residual connections, normalization, and a downprojection layer for the RNN's expanded hidden states. For more complex tasks like language modeling and Long Range Arena, standard components (convolutional layer and MLP) are added.
>
> To provide further insight into the impact of these architectural choices, we performed an ablation study on the ListOps (Long Range Arena) dataset. The results, averaged over 3 seeds, are shown below:
>
>
> |          Model         | Accuracy |
> |:----------------------:|:--------:|
> |     Random Baseline    |   0.10   |
> |         minLSTM        |   0.46   |
> |    minLSTM (+ conv)    |   0.45   |
> |     minLSTM (+ mlp)    |   0.52   |
> | minLSTM (+ conv + mlp) |   0.59   |
>
> We have updated Appendix C.1 to better explain the architectural choices and rationale behind these components.
>
> > Finally, more detailed inference speed comparisons would strengthen the paper. How do minGRU and minLSTM compare with Mamba and traditional LSTM/GRU implementations across different sequence lengths and batch sizes?
>
> Thank you for your suggestion to include more detailed inference speed comparisons. We have now included inference speed results for GRU, LSTM, minGRU, minLSTM, and Mamba (using the official implementation). However, we would like to highlight that inference speed can vary depending on hardware and implementation.
>
> For these experiments, we varied both batch size (8, 16, 32, 64) and sequence length (up to 2048). In Figure 3, we report the average inference speed across 50 runs, with context tokens considered before performing inference. Since GRU and LSTM models process context tokens sequentially, their inference times are significantly slower compared to minGRU, minLSTM, and Mamba, all of which benefit from parallel processing.
>
> Overall, minLSTM and minGRU show inference speeds comparable to Mamba. Specifically, minGRU was 6.6%, 4.1%, 4.9%, and 2.9% faster than Mamba for batch sizes of 8, 16, 32, and 64, respectively. On the other hand, minLSTM was slightly slower than Mamba by 3.6%, 2.9%, 0%, and 1.3% for the same batch sizes.
>
> Since minLSTM and minGRU are simplifications of LSTM and GRU, we expect them to generally perform faster, including during inference. This is demonstrated in Figure 4, where we compare the inference speed of minLSTM and minGRU with traditional LSTM and GRU models across varying batch sizes. As expected, minGRU and minLSTM are 19.6% and 41.5% faster than GRU and LSTM for a batch size of 64, respectively.
>
> These detailed inference speed results and plots are included in the Appendix.
>
>
> ---
>
> Martin, Eric and Chris Cundy "Parallelizing Linear Recurrent Neural Nets over Sequence Length." ICLR (2018).
>
> Gu, Albert and Tri Dao. "Mamba: Linear-Time Sequence Modeling with Selective State Spaces."  arXiv preprint arXiv:2312.00752 (2023).
>
> Tay, Yi, et al. "Long Range Arena: A Benchmark for Efficient Transformers." ICLR (2021).
>
> Delétang, Grégoire, et al. "Neural Networks and the Chomsky Hierarchy." ICLR (2023).

---

> ### Author Response · Authors · 2024-11-25
>
> Dear Reviewer jHLY,
>
> As the discussion period comes to a close, we wanted to check-in. We have addressed your concerns in the rebuttal, notably:
>
>  - **Comparison with Martin & Cundy (2018)**: We have clarified the differences between our work and theirs, emphasizing our novel contributions in the context of the recent trend toward increasingly complex sequence modeling approaches. Our work offers a counterpoint to this trend, focusing on simplicity while maintaining competitive performance.
>  - **Datasets**: We have added results from the Chomsky Hierarchy and Long Range Arena benchmarks, demonstrating that Minimal RNNs achieve performance comparable to modern state-of-the-art recurrent models, including on tasks with sequences of 1000+ steps (as you requested).
>  - **Architectural Ablation**:  We have included an ablation study of our architectural components to better illustrate the impact of each design choice.
>  - **Inference speed comparisons**: We have provided inference speed comparisons, highlighting the efficiency of Minimal RNNs, which results from their simplicity.
>
> If you have any further questions or would like additional clarification, we would be happy to address them.
>
> Thank you again for your feedback and for helping improve the manuscript.

---

> > ### Comment · Reviewer_jHLY · 2024-11-27
> > **Response**
> >
> > Hi,
> > Thanks for your response. I really appreciate all the work you have done over the course of the rebuttal, both in providing additional experiments and in explaining and clarifying your approach and updating the paper. I think the paper is much improved with your additions.
> > I would also like to thank the comments of Rui-Jie Zhu and Songlin Yang for their perspective.
> >
> > However, I am going to keep my score at 3.
> >
> > The main reason for this is the ultimate lack of novelty of the contributions. This has been extensively litigated in the author discussion responses, so I will not recap it too much. As the authors say, the main contribution of the paper is to "demonstrate that simplifying traditional, decades-old RNN architectures can yield results comparable to modern, state-of-the-art sequence models". Unfortunately, I think it is already reasonably well-known within the sequence modelling community (particularly in the subset of the community focussing on long-horizon tasks) that linear RNNs can give comparable performance to more complex sequence models such as transformers. For your minGRU architecture, this was demonstrated back in 2018 by [Martin & Cundy] and for more complicated architectures such as LSTMs this was demonstrated most recently by mLSTM. Indeed, as pointed out by another reviewer, a survey in February was titled 'On the Resurgence of Recurrent Models for Long Sequences: Survey and Research Opportunities in the Transformer Era'. An additional paper in 2023 was titled 'Resurrecting Recurrent Neural Networks for Long Sequences', which again used linearization of the RNNs for LRA tasks. Therefore, I conclude that this contribution is not really novel.
> >
> > A second factor that I take into account is the relative weakness of the experiments. While the experiments (particularly in the updates) do show that the proposed RNNs have good performance on the LRA tasks, the only task focussing on a complex, information-rich domain with extensive representation learning needed (i.e. language modelling, speech processing/generation, image generation, dna modelling) is the tiny-shakespeare dataset, which is small compared to modern corpora. While e.g. Mamba do provide examples of the success on the selective copying task, they also provide extensive results on the pile, DNA modelling, etc. HGRN use an experiment on wikitext-103, a smaller dataset (understandable for academic budgets) but still much larger than tiny-shakespeare. Matmul-free provide a scaling law over the SlimPyjama dataset. As we can see from the tiny-shakespeare results, all the models obtain presumably the optimal loss, so any differences between the various models will only appear as we increase scale (which could be analysed via a scaling law even without going to multi-billion-token datasets). Therefore, the claim that minGRU and minLSTM are actually comparable to modern models is not really substantiated for the complex, large datasets which we deal with in ML today.
> > Finally, I am skeptical of the relevance of the Long-Range Arena tasks given results such as "Never Train from Scratch: Fair Comparison of Long-Sequence Models Requires Data-Driven Priors" (ICLR 2024) which show that LRA is often testing model aspects such as initialization slight inductive biases which are not particularly relevant in evaluating models' performance on real-world tasks requiring extensive representation learning, understanding, etc.

---

### Official Review · Reviewer_aGUN · 2024-10-31

**Soundness:** 3
**Presentation:** 3
**Contribution:** 2
**Rating:** 6
**Confidence:** 4

**Summary:**

The paper investigates simplified versions of traditional recurrent neural networks (RNNs), specifically LSTMs and GRUs, adapting them for efficient training by removing their hidden state dependencies, enabling parallel training. The proposed **minLSTM** and **minGRU** models retain the functional structure of their predecessors but omit time dependencies in their gates, reducing parameter count and improving computational efficiency. Empirical tests indicate these models achieve comparable performance to contemporary state-of-the-art sequence models, suggesting that streamlined versions of older RNN architectures may offer viable alternatives in sequence modeling tasks.

**Strengths:**

- The approach effectively repurposes older RNNs by leveraging simplifications that enable parallel training, providing an interesting contrast to complex, modern architectures.

- The proposed models significantly reduce training time, achieving speed improvements of up to 175x for sequence lengths of 512, which is a notable practical advantage.

- This work challenges the abandonment of RNNs in favor of more recent architectures and suggests potential for simpler, more interpretable models in long-sequence processing.

**Weaknesses:**

*Insufficient Comparison Context:* From the current text, it is not clear whether the computational comparisons (Figure 1) were carried on considering the fact that the proposed models potentially require more layer to achieve competitor quantitative performances on tasks (Table 1). This fact potentially skews the results if competitors require fewer layers for comparable performance.

*Limited Dataset Representativeness:* The model's evaluation primarily relies on synthetic, simplified datasets, which are not representative of those used in current literature benchmarks (see Section 7/Table 4 in  [3]). This limits insights into the model’s scalability and ability to handle real-world data complexities (and it is not clear whether this was caused by memory requirements drawbacks). Moreover, the removal of hidden state dependencies from gates raises concerns regarding the model ability to preserve long range dependendencies - something that should be investigated.

*Literature Comparisons and references:* The paper references but does not thoroughly compare against xLSTM, a similar recent work aimed at improving LSTM performance. A more detailed theoretical (at least) comparison with xLSTM would strengthen the paper's argument for minLSTM's contributions. Moreover, the paper is inspired and puts emphasis (even in the title) on the recent **resurgence** of RNNs - a concept that was highligthed by recent surveys [2,3] that could help the reader better contextualize the work.  In the aforementioned surveys, other very related works [4,5] are described that should be at least mentioned

[1] Beck, Maximilian, et al. "xLSTM: Extended Long Short-Term Memory." arXiv preprint arXiv:2405.04517 (2024).

[2] Tiezzi, Matteo, et al. "On the resurgence of recurrent models for long sequences: Survey and research opportunities in the transformer era." arXiv preprint arXiv:2402.08132 (2024).

[3] Tiezzi, Matteo, et al. "State-Space Modeling in Long Sequence Processing: A Survey on Recurrence in the Transformer Era." arXiv preprint arXiv:2406.09062 (2024).

[4] Bradbury, James, et al. "Quasi-recurrent neural networks." arXiv preprint arXiv:1611.01576 (2016).

[5] Martin, Eric, and Chris Cundy. "Parallelizing linear recurrent neural nets over sequence length." arXiv preprint arXiv:1709.04057 (2017).

**Questions:**

1. **Layer Depth in Computational Comparisons:** (Please refer to Weaknesses) Were the computational benchmarks (Figure 1) conducted with models having comparable quantitative performances? The paper could benefit from a comparison on computational complexity normalized by model performances or depth.

2. **Parameter Matching for Competitors:** Related to previous question. When taking results from other papers (as reported in the Appendix), were the competitor models matched in terms of parameters and model size? Variability in model configurations could impact the validity of performance comparisons.

3. **Choice of Datasets:** The model is mainly tested on very simple datasets (mostly synthetic) that are not very representative for the current literature (see [3]). The authors should comment on why this choice taken.  If the model cannot scale to large datasets due to its inner working and memory requirements (as reported in the Limitation section), this is something that hinder the paper contributions.  For instance, the Long Range Arena benchmark has became a standardize framework to test sequence models performances, that could help also in identifying the model ability to preserve long term dependencies -- that is something extremely requested by current literature. The removal of time dependecy questions the model ability to perform in such benchmarks, which I believe should be better investigated. Thus,  at least a discussion on this is required.

4. **Theoretical Comparison with xLSTM [1]:** Could the authors elaborate on the theoretical distinctions and advantages of minLSTM over xLSTM, especially in terms of efficiency and sequence modeling?

---

> ### Author Response · Authors · 2024-11-21
> **Response to Reviewer aGUN (1/4)**
>
> We would like to thank the reviewer for their detailed review and helpful feedback.
>
> > Insufficient Comparison Context: From the current text, it is not clear whether the computational comparisons (Figure 1) were carried on considering the fact that the proposed models potentially require more layer to achieve competitor quantitative performances on tasks (Table 1). This fact potentially skews the results if competitors require fewer layers for comparable performance. [...]
> >
> > Layer Depth in Computational Comparisons: (Please refer to Weaknesses) Were the computational benchmarks (Figure 1) conducted with models having comparable quantitative performances?
>
> Thank you for your comment. To address your concern, we would like to clarify that the computational comparisons in Figure 1 were carried out using models with the same number of layers to ensure a fair and direct comparison. Empirically, we found that the parallelizable models (Minimal RNNs and Mamba) achieved comparable performance with the same layer depth. For the various experiments presented in the paper, the number of layers was kept consistent across models when comparing their performance. The only exception is Table 1, where we varied the number of layers for analysis purposes.
>
> We have updated the manuscript to explicitly clarify that the computational comparisons (Figure 1) were conducted with models having the same number of layers.
>
>
> > Literature Comparisons and references: The paper references but does not thoroughly compare against xLSTM, a similar recent work aimed at improving LSTM performance. A more detailed theoretical (at least) comparison with xLSTM would strengthen the paper's argument for minLSTM's contributions.  [...]
> >
> > Theoretical Comparison with xLSTM [1]: Could the authors elaborate on the theoretical distinctions and advantages of minLSTM over xLSTM, especially in terms of efficiency and sequence modeling?
>
> Thank you for your insightful comments. We agree that a more detailed theoretical comparison with xLSTM would strengthen the argument for minLSTM’s contributions.
>
> Although there are similarities, our work is conceptually distinct from xLSTM in that it challenges the common assumption that increasing model complexity is necessary for achieving state-of-the-art performance. While xLSTM introduces several extensions to the traditional LSTM, such as exponential gates and additional normalization states, our approach demonstrates that minimal, simplified versions of LSTMs can perform comparably to modern state-space models. This finding encourages the community to reconsider the implicit bias toward increasing architectural complexity.
>
> To clarify the differences between xLSTM and minLSTM. xLSTM introduces two variants:
>  - sLSTM retains the hidden state, making it sequential-only and less efficient.
>  - mLSTM removes the hidden state dependencies to enable parallelization, introduces a matrix memory cell, and uses a query vector for retrieval from the memory. This makes mLSTM more efficient and parallelizable but still retains considerable complexity.
>
> minLSTM, in contrast, is a fully parallelized simplification of the standard LSTM. By removing the cell state, hidden state dependencies, and its corresponding tanh activation, minLSTM drastically reduces the model's complexity, making it both easier to implement and highly efficient, while still retaining competitive performance.
>
> In addition to the theoretical distinctions, we have included new empirical comparisons with xLSTM on the Long Range Arena (Tay et al., 2021) and Chomsky Hierarchy (Delétang et al., 2023) benchmarks (see response to Datasets), both of which are well-established and were used in xLSTM's paper for benchmarking. Empirically, minLSTM achieves comparable performance to xLSTM across all datasets, despite its simpler architecture.
>
> We have updated the manuscript to clarify these theoretical differences and emphasize the efficiency and conceptual simplicity of minLSTM relative to xLSTM.

---

> ### Author Response · Authors · 2024-11-21
> **Response to Reviewer aGUN (2/4)**
>
> > Limited Dataset Representativeness: The model's evaluation primarily relies on synthetic, simplified datasets, which are not representative of those used in current literature benchmarks (see Section 7/Table 4 in [3]). [...]
> >
> > Choice of Datasets: The model is mainly tested on very simple datasets (mostly synthetic) that are not very representative for the current literature (see [3]). [...]
> >
> > For instance, the Long Range Arena benchmark has became a standardize framework to test sequence models performances, that could help also in identifying the model ability to preserve long term dependencies -- that is something extremely requested by current literature.
>
> We appreciate your concerns about the representativeness of the datasets used in our initial evaluation. We have expanded our experiments to address this.
>
> In response to your comment, we have now included Long Range Arena and Chomsky Hierarchy benchmarks, which are recognized in the literature as robust frameworks for evaluating sequence models. Together, these benchmarks provide a more representative test of a model's ability to generalize and handle both complex and long-range dependencies, which are crucial for modern sequence modeling tasks.
>
> To ensure consistency, we retrieve results (accuracy -- higher is better) for these benchmarks from the xLSTM paper and compare minLSTM with other fully parallelizable modern sequence models (e.g., RWKV, Mamba, and xLSTM[1:0], which uses the mLSTM module). We follow the model size and hyperparameters outlined in xLSTM (e.g., model dimensions, embedding dimensions, number of layers) and report results averaged across three random seeds.
>
> Empirically, we find that minLSTM achieve comparable performance to state-of-the-art models like Mamba and xLSTM on these more challenging benchmarks. This shows that minLSTM can indeed scale to handle the long-term dependencies and real-world complexities often emphasized in current literature.
>
> We have updated the manuscript to reflect these expanded experiments and highlight the relevance of these new benchmarks in demonstrating the scalability and practical applicability of simplified versions of decades-old RNNs.
>
> |                     | Bucket Sort | Missing Duplicate | Cycle Nav. | Even Pairs | Majority | Majority Count | Retrieval | ListOps | G-Image | Average |
> |:-------------------:|:-----------:|:-----------------:|:----------:|:----------:|:--------:|:--------------:|:---------:|:-------:|:-------:|:-------:|
> | Llama (Transformer) |     0.92    |        0.08       |    0.04    |     1.0    |   0.37   |      0.13      |    0.85   |   0.38  |   0.54  |   0.48  |
> |        Mamba        |     0.69    |        0.15       |    0.86    |     1.0    |   0.69   |      0.45      |    0.90   |   0.33  |   0.69  |   0.64  |
> |        RWKV-4       |     0.54    |        0.21       |    0.13    |     1.0    |   0.63   |      0.13      |    0.90   |   0.39  |   0.69  |   0.51  |
> |        xLSTM        |     0.97    |        0.33       |    0.86    |     1.0    |   0.74   |      0.46      |    0.91   |   0.41  |   0.70  |   0.71  |
> |    minLSTM (Ours)   |     0.94    |        0.26       |    0.79    |     1.0    |   0.93   |      0.47      |    0.89   |   0.59  |   0.67  |   0.73  |

---

> ### Author Response · Authors · 2024-11-21
> **Response to Reviewer aGUN (3/4)**
>
> > limits insights into the model’s scalability and ability to handle real-world data complexities (and it is not clear whether this was caused by memory requirements drawbacks). [...]
> >
> > If the model cannot scale to large datasets due to its inner working and memory requirements (as reported in the Limitation section), this is something that hinder the paper contributions.
>
> We would like to clarify that the scalability limitations discussed in the Limitation section are due to hardware constraints, rather than any inherent limitations of the model itself. Specifically, while models like Mamba and xLSTM were run on modern A100 GPUs with 80 GB of memory, our experiments were conducted on older GPUs (e.g., P100, T4, and Quadro 5000) with only 16 GB of memory (roughly 20% of the memory available to the other models). These hardware constraints significantly impacted our ability to perform large-scale experiments, especially on long sequences or large datasets.
>
> However, to address this concern, we have expanded our experimental evaluation to include results from Long Range Arena, which is designed to test models' ability to handle complex dependencies over long sequences. Despite the simplicity of the Minimal RNN architecture, our experiments show that Minimal RNNs can effectively handle these complex tasks, achieving comparable performance to recent state-of-the-art sequence models.
>
> Additionally, in Figure 1, we provide an analysis of the memory and runtime requirements of Minimal RNNs during training with respect to sequence length. The analysis shows that the computational requirements for minGRU and minLSTM are significantly lower than those of more complex models like Mamba. This suggests that, given sufficient computational resources, Minimal RNNs are scalable with dataset size and can handle large-scale datasets without the memory and computational bottlenecks that more complex models experience.
>
> To clarify further, we have updated the manuscript to emphasize that the observed scalability limitations were primarily due to the hardware constraints and not to any fundamental limitations of the Minimal RNN architecture itself. We believe that these updates, including the new experiments and analyses, highlight the scalability and efficiency of the model, particularly in comparison to more complex sequence models.
>
>
> > Moreover, the removal of hidden state dependencies from gates raises concerns regarding the model ability to preserve long range dependendencies - something that should be investigated. [...]
> >
> > The removal of time dependecy questions the model ability to perform in such benchmarks, which I believe should be better investigated. Thus, at least a discussion on this is required.
>
> Thank you for your thoughtful comment. We understand your concern regarding the removal of hidden state dependencies and its potential impact on the model’s ability to capture long-range dependencies. In our work, the removal of these dependencies was a necessary step to enable the application of the parallel scan algorithm to LSTMs and GRUs. This approach is similar to what has been done in other recent models, such as xLSTM’s mLSTM module and Mamba, both of which also omit hidden state dependencies to allow full parallelization. Instead of explicitly modeling dependencies in the gates, these models learn long-range dependencies through stacking multiple layers.
>
> While there are theoretical concerns about the potential limitations of this approach, as discussed in works like *The Illusion of State in State-Space Models* (Merrill et al., 2024), there is substantial empirical evidence supporting the effectiveness of such models. For example, in the xLSTM paper, the fully parallelized version (xLSTM[1:0]) performs similarly to—and in some cases better than—versions that retain hidden state dependencies (e.g., xLSTM[7:1]), as shown in their Tables 1, 3, 4 and Figures 5, 6, 7, 8.
>
> Empirically, we have found that Minimal RNNs also perform comparably with these state-of-the-art models on numerous benchmarks (e.g., Long Range Arena and Selective Copying), which requires the model to handle long-range dependencies. This provides further evidence that removing hidden state dependencies does not preclude effective learning of long-range dependencies, but rather enables models to scale more efficiently.
>
> In response to your concern, we have added a discussion in the methodology section (Section 3.1.1) to more thoroughly address the theoretical limitations of removing hidden state dependencies, while emphasizing the empirical success of fully parallel sequence models in capturing long-range dependencies despite this simplification.

---

> ### Author Response · Authors · 2024-11-21
> **Response to Reviewer aGUN (4/4)**
>
> > Moreover, the paper is inspired and puts emphasis (even in the title) on the recent resurgence of RNNs - a concept that was highligthed by recent surveys [2,3] that could help the reader better contextualize the work.
> > In the aforementioned surveys, other very related works [4,5] are described that should be at least mentioned
>
> Thank you for pointing out the relevant survey papers and related works that contextualize the resurgence of RNNs. We have carefully considered your suggestions and made the following updates to the manuscript:
>
>  - Introduction: We have updated the introduction to include references to the surveys [2, 3], which provide important background on the recent resurgence of RNNs and help position our work within this broader context.
>  - Related Work Section: We have expanded the discussion on parallelizing RNNs and included a more detailed comparison to the prior approaches [4, 5]. Additionally, we have added references to the surveys [2, 3] in this section to guide readers to additional resources that discuss the resurgence of RNNs in more depth.
>
> We believe these revisions will help better contextualize our work within both the historical and ongoing research in this area, providing readers with a more comprehensive understanding of the background and motivations for our approach.
>
> > Parameter Matching for Competitors: [...] When taking results from other papers (as reported in the Appendix), were the competitor models matched in terms of parameters and model size? Variability in model configurations could impact the validity of performance comparisons.
>
> Thank you for your comment. We would like to clarify that when running our experiments, we took care to ensure that our models were comparable in terms of the number of parameters and model size. Specifically, we matched the number of layers, model dimensions (e.g., embedding size, hidden size), and other key hyperparameters to those reported in the papers from which the competitor model results were cited.
>
> To ensure transparency, we have updated the Appendix to explicitly reference the papers where these parameters were retrieved from, and we provide details on the specific configurations used for each of our models.
>
> ---
>
> Tay, Yi, et al. "Long Range Arena: A Benchmark for Efficient Transformers." ICLR (2021).
>
> Delétang, Grégoire, et al. "Neural Networks and the Chomsky Hierarchy." ICLR (2023).
>
> Merrill, William, et al. "The Illusion of State in State-Space Models." ICML (2024).

---

> ### Author Response · Authors · 2024-11-25
>
> Dear Reviewer aGUN,
>
> As the discussion period comes to a close, we wanted to check-in. We have addressed your concerns in the rebuttal, notably:
>
>  - **Layer Depth in Computational Comparisons** and **Parameter Matching for Competitors**: We have clarified that the models were compared with the same number of layers and a comparable number of parameters.
>  - **Choice of Datasets**: We have added results from the Chomsky Hierarchy and Long Range Arena benchmarks (as requested), demonstrating that Minimal RNNs achieve performance comparable to modern state-of-the-art recurrent models on these widely used datasets. Concerning the removal of time dependency in gates, we have added a discussion to address the theoretical limitations of removing these dependencies, while highlighting the recent empirical success of fully parallel sequence models in capturing long-range dependencies despite this simplification.
>  - **Literature Comparisons and References**: We have clarified the differences between our work and xLSTM, and included additional references to the aforementioned surveys and related works to better position our paper within the broader literature.
>
> If you have any further questions or would like additional clarification, we would be happy to address them.
>
> Thank you again for your feedback and for helping improve the manuscript.

---

> > ### Comment · Reviewer_aGUN · 2024-11-25
> > **Rebuttal aknowledgement**
> >
> > I thank the authors for their effort in the rebuttal. I believe that the paper improved after this phase, and I will my raise my score.
> > I have one concern regarding the new Long Range Arena results, in particular on the choice of the authors to just experiment on a subset of the benchmark (i.e. the benchmark is composed by 6 tasks, **ListOps, Text, Retrieval, Image, Pathfinder, Path-X**) while the authors reported solely results in Retrieval, ListOps, Image.

---

> ### Author Response · Authors · 2024-11-26
>
> Thank you for your reply. As we followed the experimental setup used by xLSTM, we retrieved the results directly from their work. However, xLSTM did not include results for Text and Path-X, so **we have not included these tasks in our manuscript in order to maintain a direct comparison with xLSTM**.
>
> Regarding Pathfinder, we are still training the model. This task requires significantly more training time—Pathfinder was trained for 2.8 to 6 times the number of steps as the other Long Range Arena tasks (see Table 5 of the xLSTM paper). We are actively working on finalizing these results and plan to include them in the camera-ready version of the paper.
>
> We did, however, run experiments on the Text task, achieving an average score of 0.89, which is competitive with recent recurrent sequence models such as S5, RWKV, and LRU (as shown in Table 4 of RWKV's paper and Table 8 of LRU's paper). However, as there were no direct results from xLSTM for comparison, we decided not to include these results in the updated manuscript.
>
> For Path-X, the sequences are 16 times longer than those of Pathfinder, making these tasks require significantly longer to train. We are exploring potential solutions but will not be able to include Path-X results in the current paper.

---

### Official Review · Reviewer_z59C · 2024-11-04

**Soundness:** 4
**Presentation:** 3
**Contribution:** 4
**Rating:** 8
**Confidence:** 4

**Summary:**

The paper introduces minimal Long Short-Term Memory (minLSTM) and minimal Gated Recurrent Unit (minGRU) models.
The authors modify the traditional LSTM and GRU models by removing dependencies on prior hidden states from the gating mechanisms, enabling parallelization through a method similar to approaches in linear RNNs and state-space models.
The changes to the gating mechanisms significantly alter how minLSTMs and minGRUs are expected to function when compared with LSTMs and GRUs, but it allows for efficient computation without the limitations that result from back-propagation through time.
Comparisons are made with state-of-the-art models, including the state-space model Mamba.
Empirical results demonstrate that the minLSTM and minGRU models achieve competitive performance on the selective copy task, multiple reinforcement learning tasks, and a language modeling task based on the Shakespeare dataset.

**Strengths:**

The core idea of creating minimal versions of LSTM and GRU models for efficient parallel training is compelling.
I believe this contribution is novel and could be highly useful.
The benchmark results are encouraging and the comparisons made with other models seem appropriate.

**Weaknesses:**

As the authors point out, computational restrictions prevent them from providing large-scale experiments. I would be curious to see performance on additional benchmarks, such as WikiText103, Pile or the long range arena. However, I still believe the submission is strong without them.

----

Small typo: line 370, should read '... recurrent sequence models that can *be* trained in parallel ...'

in B.1, parallel_scan_log: log_x0_plus_b_star is not defined, should this be log_h0_plus_b_star?

in B.3.1 the pseudocode takes x_t but the code uses x

For the purposes of reproducibility, providing anonymized code in the supplementary material would strengthen the submission.

**Questions:**

Under "Parameter Initializations" you state that minLSTM and minGRU are stable with default initialization, in contrast to the specialized initializations in prior linear RNN papers. Did you perform any studies using different initialization methods?

When trying to replicate minGRU using the log-space pseudocode I am struggling to get the sequential and parallel modes to match, could this pseudocode be checked through? (update: I managed to do it using the cited paper, but I think the pseudocode provided is incorrect)

---

> ### Author Response · Authors · 2024-11-21
> **Response to Reviewer z59C (1/2)**
>
> We would like to thank the reviewer for their helpful feedback and their support. We are pleased to see your enthusiasm for our work.
>
> > As the authors point out, computational restrictions prevent them from providing large-scale experiments. I would be curious to see performance on additional benchmarks, such as WikiText103, Pile or the long range arena.
>
>
> Thank you for your suggestion regarding additional benchmark comparisons. We have conducted experiments on both the Long Range Arena (Tay et al., 2021) and the Chomsky Hierarchy (Delétang et al., 2023) benchmarks, which are well-established in the literature for evaluating sequence models.
>
> For these tasks, we closely follow the hyperparameters of xLSTM (Beck et al., 2024), including model size, embedding dimensions, and number of layers. The baseline results (accuracy -- higher is better) are taken from the xLSTM paper (Figure 4 for Chomsky Hierarchy and Table 6 for Long Range Arena). We compare against other parallelizable modern sequence models such as xLSTM[1:0] (which uses their parallelizable mLSTM module).
>
> In our experiments, we focus on tasks from the Chomsky Hierarchy where models have achieved at least 30% accuracy, indicating partial solvability. Results are reported across 3 random seeds for consistency.
>
> Our experiments show that Minimal RNNs achieve competitive performance with state-of-the-art models (e.g., Mamba and xLSTM) across all tasks on these benchmarks. An extended table with additional results and discussion are included in the paper.
>
> |                     | Bucket Sort | Missing Duplicate | Cycle Nav. | Even Pairs | Majority | Majority Count | Retrieval | ListOps | G-Image | Average |
> |:-------------------:|:-----------:|:-----------------:|:----------:|:----------:|:--------:|:--------------:|:---------:|:-------:|:-------:|:-------:|
> | Llama (Transformer) |     0.92    |        0.08       |    0.04    |     1.0    |   0.37   |      0.13      |    0.85   |   0.38  |   0.54  |   0.48  |
> |        Mamba        |     0.69    |        0.15       |    0.86    |     1.0    |   0.69   |      0.45      |    0.90   |   0.33  |   0.69  |   0.64  |
> |        RWKV-4       |     0.54    |        0.21       |    0.13    |     1.0    |   0.63   |      0.13      |    0.90   |   0.39  |   0.69  |   0.51  |
> |        xLSTM        |     0.97    |        0.33       |    0.86    |     1.0    |   0.74   |      0.46      |    0.91   |   0.41  |   0.70  |   0.71  |
> |    minLSTM (Ours)   |     0.94    |        0.26       |    0.79    |     1.0    |   0.93   |      0.47      |    0.89   |   0.59  |   0.67  |   0.73  |
>
>
> > Small typo: line 370, should read '... recurrent sequence models that can be trained in parallel ...'
>
> Thank you for pointing out this typo. We have fixed it in the new version.
>
>
> > For the purposes of reproducibility, providing anonymized code in the supplementary material would strengthen the submission.
>
> We will be releasing our code alongside our camera-ready version.

---

> ### Author Response · Authors · 2024-11-21
> **Response to Reviewer z59C (2/2)**
>
> > Under "Parameter Initializations" you state that minLSTM and minGRU are stable with default initialization, in contrast to the specialized initializations in prior linear RNN papers. Did you perform any studies using different initialization methods?
>
> For the tasks considered in this paper, we did not observe any need for alternative initialization schemes beyond the default PyTorch initialization for minLSTM and minGRU.
> We did find, however, that increasing the bias of the forget gate in minLSTM improved training efficiency as it encourages the model to retain information earlier during training, which in turn sped up convergence and stabilized the model during training. We have included this plot in Figure 5 of the Appendix.
>
> However, our objective of this paper was not to achieve the absolute best performance, but rather to demonstrate that just simplifying LSTMs and GRUs can still achieve performance comparable to modern methods.
>
> We have included this initialization detail in the Appendix to clarify that initialization choices can potentially further improve the model.
>
> > in B.1, parallel_scan_log: log_x0_plus_b_star is not defined, should this be log_h0_plus_b_star?
> > in B.3.1 the pseudocode takes x_t but the code uses x [...]
> > When trying to replicate minGRU using the log-space pseudocode I am struggling to get the sequential and parallel modes to match, could this pseudocode be checked through? (update: I managed to do it using the cited paper, but I think the pseudocode provided is incorrect)
>
> Thank you for your careful review and for pointing out these issues. As you correctly noted, there were a few inconsistencies in the pseudocode. We have thoroughly reviewed and updated the manuscript to address these points, including:
>
>  - B.1: We have updated (1) `log_x0_plus_b_star` to `log_h0_plus_b_star`, (2) `a_star ... dim=-1` to `a_star ... dim=1`, and (3) `torch.exp(log_h)` to `torch.exp(log_h)[:, 1:]`
>  - B.3.1: We have updated the pseudocode to consistently use x_t instead of x.
>
> Regarding the issue with matching the sequential and parallel modes, we apologize for the confusion caused by the pseudocode. We have carefully revisited the log-space pseudocode and have made corrections to ensure the sequential and parallel modes are consistent. We believe this revision resolves the issue you encountered, but please let us know if further clarification or adjustments are needed.
>
> We greatly appreciate your attention to detail and your constructive feedback.
>
>
>
> ---
>
> Tay, Yi, et al. "Long Range Arena: A Benchmark for Efficient Transformers." ICLR (2021).
>
> Delétang, Grégoire, et al. "Neural Networks and the Chomsky Hierarchy." ICLR (2023).
>
> Beck, Maximilian, et al. "xLSTM: Extended Long Short-Term Memory." arXiv preprint 	arXiv:2405.04517 (2024).

---

> > ### Author Response · Authors · 2024-11-25
> >
> > Dear Reviewer z59C,
> >
> > Thank you for your enthusiasm and insightful feedback on our work. As the discussion period comes to a close, we wanted to check-in. In our response, we have addressed the following:
> >
> > In our response, we have included the following:
> >
> >  - **Additional Datasets**: We have added results from the Chomsky Hierarchy and Long Range Arena benchmarks (as requested), demonstrating that Minimal RNNs achieve performance comparable to modern state-of-the-art recurrent models on these tasks as well.
> >  - **Initialization Analyses**:  We have included additional results with an alternative initialization, showing that certain initializations can speed up training.
> >  - **Pseudocode**: We appreciate your careful review of the paper and for catching the issues with the pseudocode. We have updated the paper to ensure its correctness.
> >
> > If you have any further questions or need additional clarification, we would be happy to address them.
> >
> > Thank you again for your valuable feedback and for helping to improve the manuscript.

---

> > > ### Comment · Reviewer_z59C · 2024-11-26
> > >
> > > I thank the authors for the great work they have contributed during the discussion phase. I am happy to see that the relatively minor points I raised about the pseudocode have been addressed.
> > >
> > > The problems highlighted by other reviewers are not so easily addressed. It is true that important parts of the contribution have been done before. Particularly in the xLSTM paper, they remove the prior hidden state dependency from an LSTM and train the model in a parallel way. I was not aware of this at the time of my review.
> > >
> > > However, I believe my initial positive reaction to the paper is well justified. The paper is clearly written and presents interesting results. In many ways, I find the paper easier to follow than prior work. It is difficult to know how to respond in this situation. The authors need to be clear and honest about novelty issues. Nevertheless, given that I still believe this paper can contribute positively to the discussion in a fast-moving area of research, I am comfortable keeping my score as an 8.

---

### Public Comment · ~Songlin_Yang1 · 2024-11-26
**Missing reference**

Dear Authors and Reviewers,

I would like to bring to your attention that minGRU bears a strong resemblance to our prior work on Hierarchically Gated Recurrent Neural Networks (HGRN), published at NeurIPS 2023, which we noted originates from GILR (introduced at ICLR 2018 in "Parallelizing Linear Recurrent Neural Nets Over Sequence Length"). Reviewer jHLY also highlighted the connection between minGRU and GILR in their review.

 Our HGRN paper already demonstrates comprehensive results on both synthetic datasets (including LRA) and billion-scale language modeling tasks.  Reviewer rQhu also mentioned the matmul-free LM, which employed a simplified version of HGRN as the gated linear RNN layer.

Furthermore, in our follow-up work "HGRN2: Gated Linear RNNs with State Expansion" (COLM 2024), we address HGRN's state size limitations by introducing an efficient state expansion technique inspired by gated linear attention, which is also closely related to RWKV6/mLSTM/Mamba2/etc

Given these existing work in the field, we kindly request that the reviewers consider this line of work when evaluating the novelty and significance of the current submission.

Best,

Songlin

---

> ### Author Response · Authors · 2024-11-26
>
> Thank you for bringing your work to our attention. We acknowledge the similarities between **minGRU**, **HGRN**, and **GILR**, as all three leverage linear recurrent models with input-dependent gating mechanisms, where the forget and input gates sum to 1. We appreciate the opportunity to clarify these connections, and have updated the **Related Work** section of our paper to include references to **HGRN** and **HGRN2** for completeness.
>
> For the reviewers' convenience, we would like to highlight key differences and contributions of our work:
>  - **GILR** was primarily designed to augment RNNs (notably **LSTMs**) by making them parallelizable, replacing hidden state dependencies with a surrogate (GILR's output), thus improving computational efficiency.
>  - **HGRN** emphasized the importance of forget gates, which had been previously neglected in state-space models and linear attention mechanisms. **HGRN** introduces a token mixer (HGRU) and a channel mixer (GLU). The **token mixer** (HGRU) incorporates gated linear recurrent layers (similar to **GILR** and **minGRU**) but extends them with components such as complex values (polar coordinates) and an additional output gate. While this increases the potential expressiveness of the block, it also introduces additional architectural and algorithmic complexity.
>
> In contrast, our work revisits the historical evolution of sequence modeling, focusing specifically on **LSTMs** (1997) and **GRUs** (2014), which were the popular and dominant models for two decades before the rise of Transformer-based models. We demonstrate that by simplifying **LSTMs** and **GRUs**, we can achieve models that perform competitively with **Transformers**, as well as with other recent models like **Mamba** and **xLSTM**, which sought to replace LSTMs and GRUs.
>
> Our contribution, therefore, lies in **simplification** rather than augmentation. By stripping down the components of RNNs to their core elements, we challenge the prevailing trend towards increasing architectural and algorithmic complexity in the field. Furthermore, our **minGRU** and **minLSTM** implementations, written in plain PyTorch with only a few lines of code, aim to make the models more accessible and easier to extend. This contrasts with the modern trend of complex CUDA-based implementations, making **minGRU** and **minLSTM** both **lightweight and highly adaptable for beginners, practitioners, and researchers**.
>
> In summary, while there are indeed connections between our work and the **HGRN/GILR** line of research, we believe our approach offers a novel perspective by simplifying existing architectures and demonstrating that simpler models can still be competitive with more complex alternatives. We hope this helps clarify the distinctions and contributions of our work.
>
> Thank you again for your feedback and the opportunity to clarify these points.

---

> ### Public Comment · ~Songlin_Yang1 · 2024-11-26
>
> Dear authors,
>
> Thank you for your prompt clarification. I would like to highlight that the simplest GILR (Section 3.1 in the original paper) is essentially in the form of a simple GRU, while what you describe—GILR-LSTM—is an additional extension. There are numerous works aiming to simplify architectures, and HGRN itself is already a highly simplified design that can be implemented wherever a minimal GRU can be implemented. Our proposed simple enhancement has also been ablated in our HGRN paper.
>
> Best,
>
> Songlin

---

> ### Author Response · Authors · 2024-11-27
> **Response to public comment**
>
> Thank you for sharing your concerns about our paper.
>
> We chose a question-based title for our work intentionally, in order to avoid overstatement of the contributions. The title was meant to prompt a broader discussion within the research community about the evolution of sequence modeling, with a particular emphasis on revisiting simpler foundational methods like LSTM and GRU. Given the increasing complexity of newer recurrent models, we felt this perspective was timely and notable.
>
> Throughout the discussion phase, we have made multiple efforts to be transparent about the limitations of our experiments, add more experiments, and to properly cite relevant prior research, including recent contributions, some of which you have also highlighted.
>
> Additionally, we want to clarify that we, the authors, did not promote our work on social media at all. Attention-generating posts were initiated independently by reputable researchers and were beyond our control. We understand how these public discussions can influence perceptions, but we want to emphasize that we did not seek or encourage such attention.
>
> We deeply respect your dedication to advancing linear RNN research and genuinely value your input. We believe that constructive dialogue and collaboration are the most productive ways to move the field forward.

---

> ### Public Comment · ~Songlin_Yang1 · 2024-11-27
>
> Dear authors:
>
> I sincerely thank the authors for engaging in this discussion.
>
> > Given the increasing complexity of newer recurrent models, we felt this perspective was timely and notable.
>
> Thank you for your comment and for revisiting simplified LSTM and GRU architectures, which revalidates the effectiveness of our proposed HGRN model. While simplifying architecture is meaningful, I must raise several concerns. Our subsequent research has already demonstrated the limitations of a simple HGRN implementation, and we have since focused on developing more expressive linear RNNs while maintaining hardware efficiency. This work seems to step backward in the progression of linear RNN architectures since the original HGRN, making me question the value of repeating these earlier discoveries.
>
> >  Throughout the discussion phase, we have made multiple efforts to be transparent about the limitations of our experiments, add more experiments, and to properly cite relevant prior research, including recent contributions, some of which you have also highlighted.
>
> Thanks for your efforts in adding new experimental results. Unfortunately, the experimental validation still appears incomplete - while our HGRN paper provided comprehensive Long Range Arena (LRA) results, your work only presents several synthetic benchmark results  [link1](https://openreview.net/forum?id=GrmFFxGnOR&noteId=sg3B6N3wkB) and no better than our reported results on HGRN. Moreover, the lack of real-world data experiments represents a significant limitation of the paper in its current form.
>
> While I appreciate your efforts to cite relevant prior research, I must point out that your treatment of Martin and Cundy (2018) appears cursory. Furthermore, your clarification regarding GILR in [link2](https://openreview.net/forum?id=GrmFFxGnOR&noteId=yAHMnIIyGR) and [link3](https://openreview.net/forum?id=GrmFFxGnOR&noteId=GP7qPxv9xq) is *very* misleading: GILR is not an augmentation but rather a simplification of GRU, equivalent to minGRU. This distinction should be clearly stated in the main paper for accuracy and transparency. I would strongly encourage you to address these points, particularly by explicitly acknowledging that GILR = minGRU in the main paper
>
> > Additionally, we want to clarify that we, the authors, did not promote our work on social media at all. Attention-generating posts were initiated independently by reputable researchers and were beyond our control. We understand how these public discussions can influence perceptions, but we want to emphasize that we did not seek or encourage such attention.
>
> Thank you for the clarification. I hope we can maintain high academic standards and ensure accurate information reaches the broader research community to avoid potential misunderstandings.
>
> Best regards,
>
> Songlin

---

### Author Response · Authors · 2024-11-28
**Thank You for Your Feedback**

Dear All,

We would like to express our sincere thanks for the time and effort you've dedicated to reviewing our paper and contributing to the discussion. We truly appreciate your thoughtful suggestions, which have greatly helped improve the quality of the work.

In response to your feedback, we have revised the abstract, introduction, related work, and conclusion to (1) better contextualize our contributions within the broader literature on efficient recurrent sequence models, and (2) highlight our work's perspective: revisiting sequence modeling from a historical perspective, with a focus on traditional RNNs from the 1990s, inviting a reevaluation of simpler models in light of newer, more complex architectures.

Thank you once again for your valuable input.

Sincerely,

The Authors

---

### Meta-Review · Area_Chair_d5kf · 2024-12-20

**Metareview:**

This paper introduces minLSTM and minGRU, simplified variants of traditional LSTM and GRU models, designed to enable parallel training by removing hidden state dependencies. These modifications result in models that are computationally efficient and achieve competitive performance on various tasks, rivaling recent models including transformers, while using fewer parameters and being fully parallelizable during training. Another contribution of the paper comes through its framing, and the attempt to spark a discussion within the community about the evolution of sequence modeling and the importance of reassessing simpler RNNs. The reviewers had differing opinions about the value this paper adds to that discussion in light of existing literature. Additionally, some reviewers raised concerns about the paper's novelty and about the empirical evaluation, questioning scalability and whether the results would be representative of modern language modeling tasks. Altogether, I do not believe this paper is appropriate for publication at ICLR at this time.

**Additional Comments On Reviewer Discussion:**

Most of the reviewers' concerns focused on novelty and empirical evaluation. The discussion on novelty was additionally extended by several public comments, pointing to similarities with previous work, (xLSTM, HGRN, MatMul-free, GILR, etc). As an example, there were many discussion a long discussion about GILR and its similarity to minGRU. The authors acknowledge the similarity, but point out that the role of minGRU in the current work is to exemplify how architectures can be simplified without sacrificing performance, whereas GILR was primarily designed to augment RNNs. With this response, the authors imply that the main novelty in minGRU is its role in advancing their overall narrative about simplification and reassessing foundational ideas in sequence modeling. However, the novelty and importance of that narrative is itself questioned by reviewers in the discussion, with one pointing to several prior works (e.g. Martin & Cundy) that establish similar perspectives in the literature. As such, the discussion on a while did raise questions about the novelty and impact of this work.

Additionally, the discussion brought up questions about the empirical evaluation. The authors did address many of the reviewer concerns, e.g. by adding additional results on Long Range Arena. Nevertheless, questions still remained about scaling behavior. The authors rightfully point out that pushing to larger scale might require unfeasibly large amounts of compute. However, reviewers point out that many relevant comparisons in the literature have results at larger scale and it might be hard to draw successful comparisons at the current scale.

Finally, one reviewer noted their score was subject to interpretation of what constitutes contemporaneous work. For transparency, note that the final decision does not hinge on this distinction.

---

### Decision · Program_Chairs · 2025-01-22

Reject